**Supercooled liquid water clouds observed over Dome C,**
**Antarctica: temperature sensitivity and cloud radiative forcing**
**Philippe Ricaud[1], Massimo Del Guasta[2], Angelo Lupi[3], Romain Roehrig[1], Eric Bazile[1],**
**Pierre Durand[4], Jean-Luc Attié[4], Alessia Nicosia[3] and Paolo Grigioni[5]**
[1]CNRM, Université de Toulouse, Météo-France, CNRS, Toulouse, France
(philippe.ricaud@meteo.fr; romain.roehrig@meteo.fr; eric.bazile@meteo.fr)
[2]INO-CNR, Sesto Fiorentino, Italy (massimo.delguasta@ino.cnr.it)
[3]ISAC-CNR, Bologna, Italy (a.lupi@isac.cnr.it; a.nicosia@isac.cnr.it)
[4]Laboratoire d'Aérologie, Université de Toulouse, CNRS, UPS, Toulouse, France
(pierre.durand@aero.obs-mip.fr; jean-luc.attie@aero.obs-mip.fr)
[5]ENEA, Roma, Italy (paolo.grigioni@enea.it)
Correspondence: philippe.ricaud@meteo.fr
27 November 2023, Version REV03 V02
Submitted to **Atmospheric Chemistry and Physics**
**Abstract**
Clouds affect the Earth climate with an impact that depends on the cloud nature (solid/
liquid water). Although the Antarctic climate is changing rapidly, cloud observations are sparse
over Antarctica due to few ground stations and satellite observations. The Concordia station is
located on the East Antarctic Plateau (75°S, 123°E, 3233 m above mean sea level), one of the
driest and coldest places on Earth. We used observations of clouds, temperature, liquid water
and surface irradiance performed at Concordia during 4 austral summers (December 2018-
2021) to analyse the link between liquid water and temperature and its impact on surface
irradiance in the presence of supercooled liquid water (liquid water for temperature less than
0°C) clouds (SLWCs). Our analysis shows that, within SLWCs, temperature logarithmically
increases from -36.0°C to -16.0°C when liquid water path increases from 1.0 to 14.0 g m$^{-2}$. The
SLWC radiative forcing is positive and logarithmically increases from 0.0 to 70.0 W m$^{-2}$ when
liquid water path increases from 1.2 to 3.5 g m$^{-2}$. This is mainly due to the downward longwave
component that logarithmically increases from 0 to 90 W m$^{-2}$ when liquid water path increases
from 1.0 to 3.5 g m$^{-2}$. The attenuation of shortwave incoming irradiance (that can reach more
than 100 W m$^{-2}$) is almost compensated for by the upward shortwave irradiance because of high
values of surface albedo. Based on our study, we can extrapolate that, over the Antarctic
continent, SLWCs have a maximum radiative forcing rather weak over the Eastern Antarctic
Plateau (0 to 7 W m$^{-2}$) but 3 to 5 times larger over Western Antarctica (0 to 40 W m$^{-2}$),
maximizing in summer and over the Antarctic Peninsula.

## 1. Introduction

Antarctic clouds play an important role in the climate system by influencing the Earth's radiation balance, both directly at high southern latitudes and, indirectly, at the global level through complex teleconnections (Lubin et al., 1998). However, in Antarctica, ground stations are mainly located on the coast and yearlong observations of clouds and associated meteorological parameters are scarce. Meteorological analyses and satellite observations of clouds can nevertheless give some information on cloud properties suggesting that clouds vary geographically, with a fractional cloud cover ranging from about 50 to 60% around the South Pole to 80-90% near the coast (Bromwich et al., 2012; Listowski et al., 2019). In situ aircraft measurements performed mainly over the Western Antarctic Peninsula (Grosvenor et al., 2012; Lachlan-Cope et al., 2016) and nearby coastal areas (O'Shea et al., 2017) provided new insights to polar cloud modelling and highlighted sea-ice production of Cloud-Condensation Nuclei (CCN) and Ice Nucleating Particles (INPs) (see e.g. Legrand et al., 2016). Mixed-phase clouds (made of solid and liquid water) are preferably observed near the coast (Listowski et al., 2019) with larger ice crystals and water droplets (Lachlan-Cope, 2010; Lachlan-Cope et al., 2016; Grosvenor et al., 2012; O'Shea et al., 2017; Grazioli et al., 2017). Based on the raDAR/liDAR-MASK (DARDAR) spaceborne products (Listowski et al., 2019), it has been found that clouds are mainly constituted of ice above the continent. The abundance of Supercooled Liquid Water (SLW, the water staying in liquid phase below 0°C) clouds depends on temperature and liquid/ice fraction. It decreases sharply poleward, and is two to three times lower over the Eastern Antarctic Plateau than over the Western Antarctic. Furthermore, the nature and optical properties of the clouds depend on the type and concentration of CCN and INPs. Bromwich et al. (2012) mention in their review paper that CCN and INPs are of various nature and large uncertainties exist relative to their origin and abundance over Antarctica. An important point remains the inability of both research and operational weather prediction models to accurately

represent the clouds (especially SLW clouds, SLWCs) in Antarctica causing biases of several
tens W m$^{-2}$ on net surface irradiance (Listowski and Lachlan-Cope, 2017; King et al., 2006,
2015; Bromwich et al., 2013) over and beyond the Antarctic (Lawson and Gettelman, 2014;
Young et al. 2019). From year-long LIDAR observations of mixed-phase clouds at South Pole
(Lawson and Gettelman, 2014), SLWCs were shown to occur more frequently than in earlier
aircraft observations or weather model simulations, leading to biases in the surface radiation
budget estimates.

76        Liquid water in clouds may occur in supercooled form due to a relative lack of ice nuclei

for temperature greater than -39°C and less than 0°C. Very little SLW is then expected because
the ice crystals that form in this temperature range will grow at the expense of liquid droplets
(called the "Wegener-Bergeron-Findeisen" process; Wegener, 1911; Bergeron, 1928;
Findeisen, 1938; Storelvmo and Tan, 2015). Nevertheless, SLW is often observed at negative
temperatures higher than -20°C at all latitudes being a danger to aircraft since icing on the wings
and airframe can occur, reducing lift, and increasing drag and weight. As temperature decreases
to -36°C, SLW dramatically lessens, so it is highly difficult 1) to observe SLWCs and 2) to
quantify the amount of liquid water present in SLWCs. But during the Year Of Polar Prediction
(YOPP) international campaign, recent observations performed at the Dome C station in
Antarctica of two case studies in December 2018 have revealed SLWCs with temperature
between -20°C and -30°C and Liquid Water Path (LWP, the liquid water content integrated
along the vertical) between 2 to 20 g m$^{-2}$, as well as a considerable impact on the net surface
irradiance that exceeded the simulated values by 20 to 50 W m$^{-2}$ (Ricaud et al., 2020).

90        The Dome C (Concordia) station, jointly operated by French and Italian institutions in the

Eastern Antarctic Plateau (75°06'S, 123°21'E, 3233 m above mean sea level, amsl), is one of
the driest and coldest places on Earth with surface temperatures ranging from about -20°C in
summer to -70°C in winter. There are four main instruments relevant to this study that have
been routinely running for about 10 years: 1) The $H_2O$ Antarctica Microwave Stratospheric and
Tropospheric Radiometer (HAMSTRAD, Ricaud et al., 2010a) to obtain vertical profiles of
temperature and water vapour, as well as the LWP. 2) The tropospheric depolarization LIDAR
(Tomasi et al., 2015) to obtain vertical profiles of backscatter and depolarization to be used for
the detection of SLWCs. 3) An Automated Weather Station (AWS) to provide screen-level air
temperature. And 4) the Baseline Surface Radiation Network (BSRN) station to measure
downward and upward longwave (4 to 50 μm) and shortwave (0.3 to 3 μm) surface irradiances
($F$) from which the net surface irradiance ($F_{Net}$), calculated as the difference between the
downward and upward components, can be computed (Driemel et al., 2018) as:
$$F_{Net} = \left(F_{LW}^{Down} - F_{LW}^{Up}\right) + \left(F_{SW}^{Down} - F_{SW}^{Up}\right) \tag{1}$$
where $F_{LW}^{Down}$, $F_{LW}^{Up}$, $F_{SW}^{Down}$, and $F_{SW}^{Up}$ represent the downward longwave, upward longwave,
downward shortwave and upward shortwave surface irradiances, respectively.
At a given time, the impact of a cloud on the surface irradiance is estimated from the
difference between the net irradiance, in cloudy ($F_{Net,cld}$) and cloud-free ($FCF_{Net}$) conditions
to provide the so-called "cloud radiative forcing" $\Delta F_{Net}$ (e.g., Stapf et al., 2020):
$$\Delta F_{Net} = F_{Net,cld} - FCF_{Net} \tag{2}$$
A similar equation can be written for each of the four irradiances that appear in the right-hand
side of equation (1). The aim of the present study is double. Using observations performed at
Concordia, we intend to quantify the link between 1) temperature in the SLWCs and LWP and
2) SLWC radiative forcing and LWP.
The article is structured as follows. Section 2 presents the instruments during the period of
study. In section 3, we detail the methodology employed to detect the SLWCs and calculate
their cloud radiative forcing, and we present the statistical method to emphasize the relationship
between in-cloud temperature and LWP on the one hand, and cloud radiative forcing and LWP
on the other hand. The results are highlighted in section 4 and discussed in section 5, before
concluding in section 6.

**2. Instruments**
We have used the observations from 4 instruments held at the Dome C station, namely the
LIDAR instrument to classify the cloud as SLWC, the HAMSTRAD microwave radiometer to
obtain LWP and vertical profile of temperature, the AWS to obtain screen-level air temperature
and the BSRN network to measure the surface irradiances ($F_{LW}^{Down}$, $F_{LW}^{Up}$, $F_{SW}^{Down}$, and $F_{SW}^{Up}$) to
obtain $F_{Net}$.
*2.1. LIDAR*
The tropospheric depolarization LIDAR (532 nm) has been operating at Dome C since 2008
(see http://lidarmax.altervista.org/englidar/_Antarctic%20LIDAR.php). The LIDAR provides
5-min tropospheric profiles of clouds characteristics continuously, from 20 to 7000 m above
ground level (agl), with a resolution of 7.5 m. For the present study, the most relevant parameter
is the LIDAR depolarization ratio (Mishchenko et al., 2000) that is a robust indicator of non-
spherical shape for randomly oriented cloud particles. A depolarization ratio below 10% is
characteristic of SLWC, while higher values are produced by ice particles. The possible
ambiguity between SLW droplets and oriented ice plates is avoided at Dome C by operating
the LIDAR 4° off-zenith (Hogan and Illingworth, 2003).
*2.2. HAMSTRAD*
HAMSTRAD is a microwave radiometer that profiles water vapour, liquid water and
tropospheric temperature above Dome C. Measuring at both 60 GHz (oxygen molecule line
($O_2$) to deduce the temperature) and 183 GHz ($H_2O$ line), this unique, state-of-the-art
radiometer was installed on site for the first time in January 2009 (Ricaud et al., 2010a and b).
The measurements of the HAMSTRAD radiometer allow the retrieval of the vertical profiles

of water vapour and temperature from the ground to 10-km altitude with vertical resolutions of 30 to 50 m in the Planetary Boundary Layer (PBL), 100 m in the lower free troposphere and 500 m in the upper troposphere-lower stratosphere. The integral along the vertical of the water vapour concentration gives the integrated water vapour (IWV). The time resolution is adjustable and fixed at 60 seconds since 2018. Note that an automated internal calibration is performed every 12 atmospheric observations and lasts about 4 minutes. Consequently, the atmospheric time sampling is 60 seconds for a sequence of 12 profiles and a new sequence starts 4 minutes after the end of the previous one. The temporal resolution on the instrument allows for detection and analysis of atmospheric processes such as the diurnal evolution of the PBL (Ricaud et al., 2012) and the presence of clouds and diamond dust (Ricaud et al., 2017) together with SLWCs (Ricaud et al., 2020). In addition, the LWP (g m$^{-2}$) that gives the amount of liquid water integrated along the vertical can also be estimated. Observations of LWP have been performed when the instrument was installed at the Pic du Midi station (2877 amsl, France) during the calibration/validation period in 2008 prior to its set up in Antarctica in 2009 (Ricaud et al., 2010a) and during the Year Of Polar Prediction (YOPP) campaign in summer 2018-2019 (Ricaud et al., 2020). At the present time, it has not yet been possible to compare HAMSTRAD LWP retrievals with observations from other instruments, neither at the Pic du Midi nor at Dome C stations. To better evaluate its performance, the 2021-2022 and the future 2022-2023 summer campaigns are dedicated to in-situ observations of SLWCs. Comparisons with numerical weather prediction models were showing consistent amounts of LWP at Dome C when the partition function between ice and liquid water was favouring SLW for temperatures less than 0°C (Ricaud et al., 2020). Note that microwave observations at 60 and 183 GHz are not sensitive to ice crystals. This has already been discussed in Ricaud et al. (2017) when considering the study of diamond dust in Antarctica. As a consequence, possible precipitation

of ice, within or below SLW clouds, as detected by the LIDAR, does not affect the retrievals of
temperature, water vapour and liquid water.
*2.3. AWS*
An American Automated Weather Station (AWS) is installed at Concordia about 500 m
away from the station and can provide screen-level air temperature ($T_a$) every 10 minutes. Data
are freely available at https://amrc.ssec.wisc.edu/data/archiveaws.html.
*2.4. BSRN*
The BSRN sensors at Dome C are mounted at the Astroconcordia/Albedo-Rack sites, with
upward and downward looking, heated and ventilated Kipp&Zonen CM22 pyranometers and
CG4 pyrgeometers providing measurements of hemispheric downward and upward broadband
shortwave (SW, 0.3 to 3 μm) and longwave (LW, 4 to 50 μm) horizontal irradiances at the
surface, respectively. These data are used to retrieve values of net surface irradiances. All these
measurements follow the rules of acquisition, quality check and quality control of the BSRN
(Driemel et al., 2018).
*2.5. Period of study*
From the climatological study presented in Ricaud et al. (2020), the SLWCs are mainly
observed above Dome C in summer, with a higher occurrence in December than in January:
26% in December against 19% in January representing the percentage of days per month that
SLW clouds were detected during the YOPP campaign (summer 2018-2019) within the LIDAR
data for more than 12 hours per day. We have thus concentrated our analysis on December and
the 4 years: 2018-2021. Since we have to use the four data sets (LIDAR, HAMSTRAD, AWS
and BSRN) in time coincidence, the actual number of days per year and the time sampling for
each day selected in our analysis are detailed in Table 1.

**3. Methodology**

*3.1. SLWC detection*

Consistent with Ricaud et al. (2020), we use LIDAR observations to discriminate between SLW and ice in a cloud. High values of LIDAR backscatter coefficient ($\beta > 100$ $\beta_{mol}$, with $\beta_{mol}$ the molecular backscatter) associated with very low depolarization ratio (<5%) signifies the presence of an SLWC whilst high depolarization ratio (>20%) indicates the presence of an ice cloud or precipitation. Once the SLWC is detected both in time and altitude, the temperature (*T*) profile within the cloud and the LWP measured by the HAMSTRAD radiometer in time coincidence are selected together with the surface irradiances observed by the BSRN instruments.

The LIDAR profiles are interpolated along the temperature vertical grid and then according to the temperature time sampling. As a consequence, for a given time and height, we have a depolarization ratio, a backscatter value, a temperature as well as (not height-dependent) IWV and LWP values. BSRN irradiances are time interpolated to be coincident with the other parameters. So, for a given time, we have a set of BSRN irradiances ($F_{LW}^{Down}$, $F_{LW}^{Up}$, $F_{SW}^{Down}$, $F_{SW}^{Up}$ and $F_{Net}$) and an LWP. At a (time, height) point showing high backscatter signal and low depolarization, the associated parameters (temperature, LWP and irradiances) are flagged as "SLW cloud". The statistic is thus done using all the SLW-flagged points without any averaging. The temperature corresponds to the in-cloud temperature.

Figure 1 shows, as a typical example, the time evolution of the LIDAR backscatter coefficient and depolarization ratio, as well as the HAMSTRAD LWP and temperature vertical profile for the 27 December 2021. Associated with the SLWCs, the LWP values are between 1.0 and 3.0 g m$^{-2}$. The SLWCs are present over a temperature range varying from about -28.0 °C to -33.0 °C. Note the cloud present at 04:00-05:00 UTC that is not labelled as a SLWC but rather as an ice cloud (high backscatter and high depolarization signals) with no associated increase of LWP and temperature above -28.0 °C.

Figure 2 highlights the time evolution of the SLWC obtained on 27 December 2021
together with some snapshots from the HALO-CAM video camera taken with or without SLWC
on: 01:00 (no SLWC), 07:19 (SLWC), 09:00 (no SLWC), 10:14 (SLWC), 13:00 (no SLWC),
16:03 (SLWC), 18:01 (no SLWC) and 20:53 UTC (SLWC). SLWCs (high backscatter and low
depolarization signals) are clearly detected at 07:00-08:00, 10:00-11:00, 16:00-17:00, 21:00-
22:00 and 23:00-24:00 UTC over an altitude range 500 to 1000 m above ground level (agl). In
general, SLWCs observed over the station did not correspond to overcast conditions.
*3.2. Cloud Radiative Forcing*
From equation (2), one of the main difficulties in computing the cloud radiative forcing
($\Delta F_{Net}$) is to estimate $FCF_{Net}$ from its individual components, namely the cloud-free downward
longwave, upward longwave, downward shortwave and upward shortwave surface irradiances.
We performed several studies (reference irradiances measured over days when clouds are
absent, radiative transfer calculations) from which it resulted that the most robust method was
to use a parameterization of the cloud-free downward longwave and shortwave surface
irradiances widely used in the community. In Dutton et al. (2004), cloud-free downward
shortwave surface irradiance ($FCF_{SW}^{Down}$) is parameterized as:

$$FCF_{SW}^{Down} = a \, \cos(z)^b \, c^{\left(\frac{1}{\cos(z)}\right)} \tag{3}$$

where $z$ is the solar-zenith angle, and $a$, $b$, and $c$ are coefficients optimized using well-identified
cloud-free situations. In Dupont et al. (2008), cloud-free downward longwave surface
irradiance ($FCF_{LW}^{Down}$) is parameterized as:

$$FCF_{LW}^{Down} = \varepsilon_a \, \sigma \, T_a^4 \tag{4}$$

where $T_a$ is the screen-level air temperature in Kelvin (K), $\sigma$ the Stephan-Boltzmann's constant
and $\varepsilon_a$ the apparent atmospheric emissivity. The latter is supposed to be a function of the
integrated water vapor (IWV) following the equation:

$$\varepsilon_a = 1 - (1 + IWV) \exp(-(d + e \times IWV)^f) \tag{5}$$

where $d$, $e$ and $f$ are coefficients that need to be optimized using cloud-free situations and IWV

is provided by the HAMSTRAD measurements. The cloud-free upward shortwave surface

irradiance ($FCF_{SW}^{Up}$) is evaluated from $FCF_{SW}^{Down}$ with the surface albedo ($A_{BSRN} = F_{SW}^{Up}(BSRN)/F_{SW}^{Down}(BSRN)$) calculated from observations:

$$FCF_{SW}^{Up} = A_{BSRN} \times FCF_{SW}^{Down} \tag{6}$$

where $F_{SW}^{Up}(BSRN)$ and $F_{SW}^{Down}(BSRN)$ are the upward and downward shortwave surface

irradiance measured by the BSRN instruments, respectively. With this method, we take into

account the actual shape of the surface, and in particular its rough structure caused by the

sastrugi (see section 5.5). Thus, the surface albedo varies with the sun angles (azimutal and

zenithal) and cannot be considered as constant over the diurnal cycle.

The cloud-free upward longwave radiation ($FCF_{LW}^{Up}$) is evaluated as:

$$FCF_{LW}^{Up} = \varepsilon_s \; \sigma \, T_s^4 \; + \; (1 - \varepsilon_s) \, FCF_{LW}^{Down} \tag{7}$$

where $T_s$ is the surface temperature and the surface emissivity $\varepsilon_s$ is assumed constant and equal

to 0.99. $T_s$ is diagnosed based on equation (7) by using the BRSN upward and downward

longwave surface irradiances.

Cloud-free situations are detected based on visual inspection of the LIDAR

(depolarization) measurements. Depolarization ratios greater than about 1% are attributed to

the presence of cloud (cirrus, mixed-phase, SLW), diamond dust, fog, etc. Thus, within each

24-hour slot covering the Decembers 2018-2021, the 1-hour periods when the depolarization

ratios are less than 1% are considered as cloud-free periods. Consequently, to evaluate the

surface cloud-free irradiances over the month of December and the years 2018-2021, we need

to have coincident observations from the 4 BSRN instruments, the LIDAR (depolarization),

HAMSTRAD and the AWS (see Table 1).

Once cloud-free situations are identified, the parametric coefficients *a-f* are estimated

minimizing a least-square cost function using the trust region reflective method (e.g., Branch

et al., 1999). To assess the robustness of the estimated coefficient values, a K-fold cross-
validation is performed. The learning dataset is split into 10 subsamples of equal size. Nine of
them are selected to optimize the coefficient and the validation is conducted on the remaining
subsample. The exercise is performed 10 times. The results are summarized below. Note that
following Dupont et al. (2008), $f$ is assumed to be equal to 1.0, and therefore not optimized.

272         For cloud-free downward shortwave surface irradiance, the K-fold cross-validation

provides the following K-fold average value (K-fold minimum and maximum are indicated
within brackets): a = 1360.7 [1360.5, 1360.8] W m$^{-2}$; b = 0.990 [0.989, 0.991]; c = 0.964 [0.964,
0.965] giving a bias of -0.002 [-0.317, 0.251] W m$^{-2}$ and a RMSE of 14.9 [10.8, 16.5] W m$^{-2}$.
Similarly, for cloud-free downward longwave surface irradiance, the K-fold cross-validation
provides the following results: d = 0.723 [0.722, 0.724]; e = 3.58 [3.57, 3.59] kg$^{-1}$ m$^{2}$; f = 1.0
giving a bias of 0.34 [-0.005, 0.87] W m$^{-2}$ and a RMSE of 9.26 [8.92, 9.58] W m$^{-2}$. These
coefficient values are then used to compute cloud-free surface irradiances at a 1-min time
resolution.

281         Figure 3 shows the time evolution of the cloud radiative forcing ($\Delta F_{net}$) and the individual

components ($\Delta F_{LW}^{Down}$, $\Delta F_{LW}^{Up}$, $\Delta F_{SW}^{Down}$ and $\Delta F_{SW}^{Up}$) calculated for 27 December 2021 when
SLWCs are present (see Figures 1 and 2). Associated with the SLWCs, on the one hand,
$\Delta F_{LW}^{Down}$ increases to values of 40 to 90 W m$^{-2}$, whilst the impact on $\Delta F_{LW}^{Up}$ is negligible (±2 W
m$^{-2}$). On the other hand, $\Delta F_{SW}^{Down}$ and $\Delta F_{SW}^{Up}$ both similarly decrease by 80 to 150 W m$^{-2}$. The
effect on $\Delta F_{net}$ is obviously positive (0 to 80 W m$^{-2}$) with some weak negative values (from 0
to -10 W m$^{-2}$) when SWLCs just appear or disappear that can possibly come from the
inhomogeneity of the cloud distribution. Spikes can be attributed to cloud edge effects, when a
fraction of the direct shortwave incident radiation and an additional diffuse contribution
scattered from cloud edges fall on the radiation sensor.
We now want to statistically analyse all the $\Delta F$ calculated in December 2018-2021 in order
to assess the SLWC radiative forcing as a function of LWP and to investigate the sensitivity of
the temperature inside the SLWCs as a function of LWP.
*3.3. Statistical Method*
The datasets corresponding to SLWCs periods are binned into 1°C-wide bins for in-cloud
temperature $T$, 0.2 g m$^{-2}$-wide bins for LWP, and 5 W m$^{-2}$-wide bins for $\Delta F$. The number of
points per bin is calculated for all the paired datasets, namely $T$-LWP, and $\Delta F$-LWP ($\Delta F_{net}$-
LWP, $\Delta F_{LW}^{Down}$-LWP, $\Delta F_{LW}^{Up}$-LWP, $\Delta F_{SW}^{Down}$-LWP and $\Delta F_{SW}^{Up}$-LWP). The 2D probability density
(PD) is calculated for the paired datasets and defined as $PD_{ij} = 100 \frac{N_{ij}}{N_t}$, where $N_{ij}$ and $N_t$ are
the count number in the bin $ij$ and the total count number ($N_t = \sum_{j=1}^{N} \sum_{i=1}^{M} N_{ij}$ ), respectively,
with $M$ and $N$ being the total number of bins in LWP on one side, and in temperature or $\Delta F$ on
the other side, respectively. So, for each value of $T_j$ (within a 1°C-wide bin $j$) or $\Delta F_j$ (within a
5 W m$^{-2}$-wide bin $j$), a weighted average of LWP ($\overline{LWP_j}$) is calculated together with its
associated weighted standard deviation ($\sigma_{LWP_j}$), considering all the $LWP_{ij}$ values (within 0.2 g
m$^{-2}$-wide bins) from $i$=1 to $M$, with $M$ the total number of LWP bins and $w_{ij}$ the weight, namely
the number of points ($w_{ij} = N_{ij}$), associated to the bin $ij$:
$$\overline{LWP_j} = \frac{\sum_{i=1}^{M} w_{ij} \, LWP_{ij}}{\sum_{i=1}^{M} w_{ij}} \tag{8}$$

and
$$\sigma_{LWP_j} = \sqrt{\frac{\sum_{i=1}^{M} w_{ij} \, (LWP_{ij} - \overline{LWP_j})^2}{\sum_{i=1}^{M} w_{ij}}} \tag{9}$$

For each $T$ and $\Delta F$ dataset, the distribution of the total count numbers $N_{tj}$ per 1°C or
5 W m$^{-2}$-wide bin ($N_{tj} = \sum_{i=1}^{M} N_{ij}$ with $j = 1, \dots, N$) can be fitted by a function $N(x)$, with $x =$
$T$ or $\Delta F$, based on 2 to 3 Gaussian distributions as:

$$N(x) = \sum_{k=1}^{2 \text{ or } 3} a_k \exp\left(-\frac{1}{2}\left(\frac{x-\mu_k}{\sigma_k}\right)^2\right) + c_0 \tag{10}$$


with $a_k$, $\mu_k$ and $\sigma_k$ being the amplitude, the mean and the standard deviation of the $k^{\text{th}}$ Gaussian
function and $c_0$ is a constant. We have used 0, 2 or 3 Gaussians for $\Delta F$ components and 3
Gaussians for $T$ ("0" means that no Gaussian fit was meaningful). Table 2 lists all the fitted
parameters ($a_k$, $\mu_k$, $\sigma_k$ and $c_0$ with $k = 0$ to 3).
In the relationship between $x$ ($T$ or $\Delta F$) and LWP, we have considered $x_j$ ($T_j$ or $\Delta F_j$) to be
significant when:

$$|x_j - \mu_k| \leq \sigma_k \text{ for } k = 1 - 2 \text{ or } 3 \text{ (for } \Delta F) \text{ or } 1 - 3 \text{ (for } T) \tag{11}$$


and used for this significant point its average value and standard deviation, $\overline{LWP_j}$ and $\sigma_{LWP_j}$,
respectively, with $j = 1, \ldots, N$.
Finally, a logarithmic function of the form

$$x = \alpha + \beta \ln(\overline{LWP}) \tag{12}$$


has been fitted onto these significant points where the retrieved constants $\alpha$ and $\beta$ are shown in
Table 3 for $x$ being $T$, $\Delta F_{net}$, $\Delta F_{LW}^{Down}$, $\Delta F_{LW}^{Up}$, $\Delta F_{SW}^{Down}$ and $\Delta F_{SW}^{Up}$.

**4. Results**
*4.1. Temperature-Liquid Water Relationship in SLWCs*
The relationship between temperature and LWP within SLWCs over the 4-summer period
at Dome C is presented Figure 4 left in the form of a Probability Density (PD) that is the fraction
of points within each bin of 0.2 g m$^{-2}$ width in LWP and 1.0°C width in temperature. It clearly
shows a net tendency for liquid water to increase with temperature, up to ~14 g m$^{-2}$ in LWP and
-18°C in temperature, with two zones having a density as high as ~2%, at [0.5 g m$^{-2}$, -33°C]
and [1.5 g m$^{-2}$, -32°C]. We have performed a weighted average of the LWPs within each
temperature bin (Figure 4 centre). Then, we have fitted 3 Gaussian distributions to the count
numbers as a function of temperature (Figure 4 right). If we now only consider temperature
bins within one-sigma of the centre of the Gaussian distributions, we can fit the following
logarithmic relation of the temperature $T$ as a function of LWP within the SLWC (Figure 4
centre):
$$T(LWP) = -33.8\,(\pm1.5) + 6.5\ln(LWP) \qquad (13)$$
for $T \in [-36; -16]$ °C and $LWP \in [1.0; 14.0]$ g m$^{-2}$, with a validity range indicated by the
2 blue dashed lines ($\pm1.5$ °C) in Figure 4 centre. In other words, based on our study, we have
a clear evidence that supercooled liquid water content exponentially increases with temperature.
Considering the temperature vs. LWP relationship, the two main Gaussian distributions are
centered around -28°C and -30°C, corresponding to temperatures usually encountered in
Concordia whilst the third one, far much less intense, is centered around -18°C, probably the
signature of very unusual events occurring in Concordia as the warm-moist events. Episodes of
warm-moist intrusions exist above Concordia originated from mid-latitudes (Ricaud et al., 2017
and 2020) and are known as "atmospheric rivers" (Wille et al., 2019). Although they are
infrequent, they can provide high values of temperature and LWP.
*4.2. Radiative Forcing-Liquid Water Relationship in SLWC conditions*

353        Although the amount of LWP is very low ($< 20$ g m$^{-2}$) at Dome C compared to what can

be measured and modelled (Lemus et al., 1997) in the Arctic (50 to 75 g m$^{-2}$) and at
middle/tropical latitudes (100 to 150 g m$^{-2}$), we intended to estimate its impact on the cloud
radiative forcing at Dome C. In Figures 5 to 9, the left panel presents the PDs of the cloud
radiative forcing $\Delta F_{net}$ as a function of the LWP, and for the individual components that
contribute to the cloud radiative forcing: $\Delta F_{LW}^{Down}$, $\Delta F_{LW}^{Up}$, $\Delta F_{SW}^{Down}$ and $\Delta F_{SW}^{Up}$, respectively. The
central panel shows, for the same parameters, the corresponding weighted average LWP within
5 W m$^{-2}$-wide bins of $\Delta F$ whereas the right panel shows the corresponding count number within
5 W m$^{-2}$-wide bins fitted by 2 or 3 Gaussian distributions (or no Gaussian distribution when it
becomes impossible).
Based on our analysis, the relationship between $\Delta F_{net}$ (W m$^{-2}$) and the LWP (g m$^{-2}$) has
been estimated as:
$$\Delta F_{net}(LWP) = -18.0\ (\pm 10.0)\ + 70.0\ln(LWP) \tag{14}$$
for $\Delta F_{net} \in [0; 70]$ W m$^{-2}$ and $LWP \in [1.2; 3.0]$ g m$^{-2}$, with a validity range indicated the two
blue dashed lines ($\pm 10.0$ W m$^{-2}$) in Figure 5 centre. Thus, for LWP greater than 1.2 g m$^{-2}$, our
study clearly shows that the cloud radiative forcing induced by the presence of SLWCs above
Concordia is positive and can reach 70 W m$^{-2}$ for an LWP of 3.0 g m$^{-2}$.
The splitting of the cloud radiative forcing between each of its four components can be
evaluated from their individual relationships with the LWP. These relations are gathered in
Table 3, established from the plots presented in Figures 5 to 9. They are of the same form as
for cloud radiative forcing, i.e. a logarithmic dependence on LWP. Table 3 presents the
coefficients $\alpha$ and $\beta$ of the logarithmic function $f(LWP) = \alpha + \beta\ln(LWP)$ for the temperature
$T$ or the radiation components $\Delta F$, together with the valid range of these relations for $T, \Delta F$ and
LWP. For the values presented in Table 3, our study clearly shows that SLWCs have a positive
impact on $\Delta F_{LW}^{Down}$ increasing from 0 to 90 W m$^{-2}$ for LWP ranging from 1.0 to 3.5 g m$^{-2}$, a
negative impact on $\Delta F_{SW}^{Down}$ and $\Delta F_{SW}^{Up}$ decreasing from 0 to -130 and -110 W m$^{-2}$, respectively
for LWP ranging from 1.5 to 4.0 g m$^{-2}$, and negligible impact ($\pm 5$ W m$^{-2}$) on $\Delta F_{LW}^{Up}$ for LWP
ranging from 0 to 6.5 g m$^{-2}$. Considering the absolute values of $\Delta F$ vs. LWP relationship
(keeping aside $\Delta F_{LW}^{Up}$), we have systematically the most intense Gaussian distributions centered
at ~10 W m$^{-2}$, and the other ones centered at ~55 W m$^{-2}$ and ~80 W m$^{-2}$.
To synthetize, our study showed that the major impact of SLWCs on net surface irradiance
is an increase of downward longwave component (0 to 80 W m$^{-2}$), whereas it has a marginal
impact on upward longwave component since this parameter is mainly dependent on $T_s$ which
results from various meteorological forcings. In the presence of SLWC, the attenuation of
shortwave incoming irradiance (which can overpass 100 W m$^{-2}$) is almost compensated for by
the upward shortwave irradiance because of high values of surface albedo.
We can also estimate the sensitivity of the longwave component to temperature and
humidity by considering the values of the equivalent atmospheric emissivity $\varepsilon_a$ used in the
equations 4-7. On the one side, the values of IWV observed at Dome C are very low even in
summer, typical summertime values are between 0.8 and 1.2 kg m$^{-2}$ (Ricaud et al., 2020). This
corresponds to values of $\varepsilon_a$ between 0.950 and 0.985, i.e. a relative variation of the order of
3.6%. On the other side, a variation $\Delta T$ of the screen-level air (surface) temperature $T_a$ ($T_s$) has
a relative impact on the downwelling (upwelling) longwave irradiance of the order of 4 $\Delta T/T_a$
(4 $\Delta T/T_s$), which amounts to around 1.6% per degree of $\Delta T$. Given that observations of surface
and screen-level air temperatures reveal variations of several degrees, both in their diurnal cycle
and from a day to another, we can conclude that the impact of temperature on longwave
irradiance variations is larger than that of IWV.

**5. Discussion**
*5.1 Relation with critical temperature*
Our study shows that, above Concordia, there is an exponential dependence of LWP on both
temperature and cloud radiative forcing, that is to say supercooled liquid water exponentially
increases with temperature in the range -36°C to -16°C. This is in agreement with the outputs
from a simple model for thermodynamic properties of water from sub-zero temperatures up to
+100°C (Sippola and Taskinen, 2018). The model shows that the density $\rho$ (g cm$^{-3}$) of liquid
water exponentially increases with temperature from -34°C to 0°C through the following
relationship:
$$\rho = \rho_0 \exp\left\{-T_c\left(A + B\varepsilon_0 + 2C\varepsilon_0^{1/2}\right)\right\}$$ (15)
where $\rho_0 = 1.007853$ g cm$^{-3}$, $A = 3.9744$ $10^{-4}$ K$^{-1}$, $B = 1.6785$ $10^{-3}$ K$^{-1}$, and $C = -7.8165$ $10^{-4}$ K$^{-1}$
$^1$; $T_c$ is the critical temperature (K) and $\varepsilon_0$ (unitless) is defined as:
$$\varepsilon_0 = \frac{T}{T_c} - 1 \tag{16}$$

where $T$ is temperature in K. In thermodynamics, a critical point is the end point of a phase
equilibrium curve. In our study, the liquid–ice boundary terminates at some critical temperature
$T_c$. $T_c$ is about 224.8 K if water is pure and free of nucleation nuclei. Sippola and Taskinen
(2018) reviewed a value of $T_c$ ~227-228 K (approx. -45°C) in the literature. This is also in
agreement with the results from our study showing that, above Concordia, we could not
observed SLWCs at temperatures less than -36°C consistent with the fact that the threshold
temperature to get SLWCs should be around -39°C (see the discussions on errors in section

421      5.3).

*5.2. Modelling SLWC*
Previous studies have already underlined the difficulty to model the SLWC together with
its impact on surface radiations. Modelling SLWCs over Antarctica is challenging because 1)
operational observations are scarce since the majority of meteorological radiosondes are
released from ground stations located at the coast and very few of them are maintained all year
long, and satellite observations are limited to 60°S in geostationary orbit whilst, in a polar orbit,
the number of available orbits does not exceed 15 per day, and 2) the model should provide a
partition function favouring liquid water at the expense of ice for temperatures between -36°C
and 0°C in order to calculate realistic SLW contents. Differences of 20 to 50 W m$^{-2}$ in the net
surface irradiance were found in the Arpege model (Pailleux et al., 2015) between clouds made
of ice or liquid water during the summer 2018-2019 (Ricaud et al., 2020), differences that are
very consistent with the results obtained in the present study. Although SLWCs are less present
over the Antarctic Plateau than over the coastal region, their radiative impact is not negligible
and should be taken into account with great care in order to estimate the radiative budget of the
Antarctic continent in one hand, and, on the other hand, over the entire Earth.
*5.3. Errors*
Measurements of temperature, LWP, depolarization signal and surface irradiances $F$ are
altered by random and systematic errors that may affect the relationships we have obtained
between LWP and either temperature or cloud radiative forcing $\Delta F_{net}$ and its individual
components. The temperature measured by HAMSTRAD below 1 km has been evaluated
against radiosonde coincident observations from 2009 to 2014 (Ricaud et al., 2015) and the
resulting bias is 0 to 2°C below 100 m and between -2 and 0°C between 100 and 1000 m.
SLWCs are usually located around 400-600 m above the ground where the cold bias can be
estimated to be about -1.0°C. The one-sigma (1-$\sigma$) RMS temperature error over a 7-min
integration time is 0.25°C in the PBL and 0.5°C in the free troposphere (Ricaud et al., 2015).
As a consequence, given the number of points used in the statistical analysis (>1000), the
random error on the weighted-average temperature is negligible (<0.02°C). The LWP random
and systematic errors are difficult to evaluate since there is no coincident external data to
compare with. Nevertheless, the 1-$\sigma$ RMS error over a 7-min integration time can be estimated
to be 0.25 g m$^{-2}$ giving a random error on the weighted average LWP less than 0.08 g m$^{-2}$. Based
on clear-sky observations, the positive bias can be estimated to be of the order of 0.4 g m$^{-2}$.
Theoretically, SLW should not exist at temperatures less than -39°C although it has been
observed in recent laboratory measurements down to -42.55°C (Goy et al., 2018). Using
equation (13) with an LWP bias of 0.4 g m$^{-2}$ gives a temperature of -39.8°C (~0.8°C lower than
the theoretical limit of -39°C), so the biases estimated for temperature and LWP are very
consistent with theory.
The estimation of systematic and random errors on LIDAR backscattering and
depolarization signals and their impact on the attribution/selection of SLWC is not trivial. But
the most important point is to evaluate whether the observed cloud is constituted of purely liquid
or mixed-phase water. Even considering the backscatter intensity only, we could not exclude
that ice particles could have been present in the SLWC events investigated in 2018 (Ricaud et
al., 2020). Therefore, in the present analysis, although we made a great attention to diagnose
ice in the LIDAR cloud observations, we cannot totally exclude ice particles thus mixed-phase
parcels were actually present when we labelled the observed cloud as SLWCs.
The 4 instruments providing $F_{LW}^{Down}$, $F_{LW}^{Up}$, $F_{SW}^{Down}$, and $F_{SW}^{Up}$ follow the rules of acquisition,
quality check and quality control of the BSRN (Driemel et al., 2018). These data are often
considered as a reference against which products based on satellite observations and radiative
transfer models (such as e.g. CERES) are validated (Kratz et al., 2020). In polar regions
(Lanconelli et al., 2011), $F_{SW}^{Down}$ and $F_{SW}^{Up}$ are expected to be affected by random errors up to
±20 W m$^{-2}$ while $F_{LW}^{Down}$ are expected to be affected by random errors not greater than ±10 W
m$^{-2}$ (Ohmura et al., 1998). As a consequence, given the large number of observations used per
5 W m$^{-2}$-wide bins (1000-3000), the random error on the weighted-average $F$ is negligible (0.3
to 0.7 W m$^{-2}$) whatever the radiations considered, LW and SW.
Finally, another source of error comes from 1) the geometry of observation and 2) the
discontinuous SLWC layer. Firstly, LIDAR is almost zenith pointing, HAMSTRAD makes a
scan in the East direction (from 10° elevation to zenith), whilst the BSRN radiometers detect
the radiation in a 2π-steradians field of view (3D configuration). That is to say, in our analysis,
the whole sky contributes to the radiation whilst only the cloud at zenith (1D configuration) and
on the East direction (2D configuration) is observed by the LIDAR and HAMSTRAD,
respectively. Secondly, SLWCs cannot be considered as uniform in the whole (see e.g. broken
cloud fields in Figure 2).
*5.4. Other clouds*
Although the method we have developed to select the SLWCs has been validated using the
amount of LWP and, in another study, using space-borne observations (Ricaud et al., 2020), we
cannot rule out that, associated with the SLW droplets, are also ice particles, that is clouds are
constituted of a mixture of liquid and solid water. Statistics of ice and mixed-phase clouds over
the Antarctic Plateau have been performed by Cossich et al. (2021) revealing mean annual
occurrences of 72.3 %, 24.9 %, and 2.7 % for clear sky, ice clouds, and mixed-phase clouds,
respectively. Generally, mixed-phase clouds are a superposition of a lower layer being made of
liquid water and an upper layer being made of solid water (see Fig. 12.3 from Lamb and
Verlinde, 2011). These mixed-layer clouds do not significantly modify the relationship between
temperature and LWP because 1) SLW observations from HAMSTRAD are only sensitive to
water in liquid phase and 2) temperature from HAMSTRAD is selected at times and vertical
heights where the LIDAR depolarization signal is very low (<5%). Although we have verified
that pure ice clouds were not selected by our method, we cannot differentiate mixed-phase
clouds from purely SLWCs.
Furthermore, we already have noticed that SLWCs developed at the top of the PBL (Ricaud
et al., 2020) in the "entrainment zone" and maintained in the "capping inversion zone",
following the terminology of Stull (1988), at a height ranging from 100 to 1000 m above ground
level. Nevertheless, at 00:00-06:00 LT when the sun is at low elevation above the horizon (24-
h polar day), the PBL may collapse down to a very low height ranging 20-50 m. In this
configuration, it is hard to differentiate from LIDAR observations between a SLWC and a fog
episode, although the LIDAR can measure depolarization (but not backscatter) down to
approximately 10-30 m above the ground (Figure S3 in Chen et al., 2017), so that we can
distinguish liquid/frozen clouds very close to the ground.
Finally, we cannot rule out that, above the SLWCs that are actually observed by both
LIDAR and HAMSTRAD, other clouds might be present, as e.g. cirrus clouds constituted of
ice crystals. These mid-to-upper tropospheric clouds cannot be detected by HAMSTRAD (no
sensitivity to ice crystals). In the presence of SLWCs either low in altitude or optically thick,
the LIDAR backscatter signal is decreased in order to avoid saturation and the signal from upper
layers is thus almost cancelled. These mid-to-high-altitude clouds are sensed by the BSRN
instruments and surface irradiance can be affected in this configuration. Based on the presence
of cirrus clouds before or after the SLWCs (and sometimes during the SLWCs if optically thin),
we can estimate that the number of days when SLWCs and cirrus clouds are simultaneously
present to cover less than 10% of our period of interest.
*5.5. Sastrugi effect on the surface albedo*
Sastrugi are features formed by erosion of snow by wind. They are found in polar regions,
and in snowy, wind-swept areas of temperate regions, such as frozen lakes or mountain ridges.
Sastrugi are distinguished by upwind-facing points, resembling anvils, which move downwind
as the surface erodes.
Figure 10 shows the BSRN surface albedo averaged over the five cloud-free days (2 and
19 December 2018; 3, 17 and 26 December 2021) showing a clear diurnal signal with a
maximum of 0.85 from 10:00 to 14:00 UTC (from 18:00 to 22:00 LT) and a minimum of 0.70
from 19:00 to 23:00 UTC (from 03:00 to 07:00 LT). The large diurnal signal present in the
observed surface albedo is likely the signature of 1) the sastrugi orientation and also 2) the sun
zenith angle which impacts on the surface albedo even with a flat snow surface (Gardner and
Sharp, 2010). Note that the surface albedo of snow under cloudy conditions may differ from
the surface albedo under cloud-free conditions (e.g., Gardner and Sharp, 2010; Stapf et al.,
2020). The BSRN $F_{SW}^{Up}$ sensor has a circular footprint. For a sensor installed at a height $h$ above
the ground, 90% of the signal comes from an area at the surface closer than 3.1 $h$ (Kassianov et
al., 2014). Since at Dome-C the instrument is installed at a height of 2-3 m, the albedo is thus
determined by the surface elements in the immediate vicinity (a few meters) of the sensor.
We have fitted the averaged cloud-free BSRN surface albedo with the sum of two sine
functions, imposing periods of 24 and 12 hours (Figure 10) together with the residuals between
the averaged surface albedo and the fitted function. We can state that the sastrugi effect on the
observed cloud-free surface albedo at Concordia is successfully fitted by two sine functions of
24h and 12h periods to within 0.003 mean absolute error, with a coefficient of determination
$R^2$ equal to 0.993 and a root mean square error of 0.0004.
Moreover, we have considered all the BSRN observations in Decembers 2018, 2019, 2020
and 2021 to calculate the albedo (Figure 11), and we have superimposed the fitted trigonometric
function as described in Figure 10. The presence of clouds is well highlighted by observations
that depart from the fitted function whilst, during periods of clear-sky conditions, BSRN
albedos coincide well with the fitted function. To conclude, the surface albedo at Concordia
should be treated considering sastrugi effect.
*5.6. Maximum SLWC Radiative Forcing over Antarctica*
Based on 2007-2010 reanalyses, observations and climate models (Lenaerts et al., 2017),
LWP over Antarctica is on average less than 10 g m$^{-2}$, with slightly larger values in summer
than in winter by 2 to 5 g m$^{-2}$. Over Western Antarctica, LWPs are larger (20 to 40 g m$^{-2}$) than
over Eastern Antarctica (0 to 10 g m$^{-2}$). As a consequence, LWPs observed at Concordia are
consistent with values observed over the Eastern Plateau, with a factor 2 to 4 smaller than those
observed over the Western continent. Based on our results and on the observed cloud fraction
($\eta_{CF}$) of SLWCs over Antarctica for different seasons (Listowski et al., 2019), we can estimate
the maximum SLWC radiative forcing at the scale of the Antarctic continent ($\Delta F_{Net-Ant}^{max}$) from
the maximum of $\Delta F_{net}$ ($\Delta F_{Net}^{max} = 70$ W m$^{-2}$) computed in our study:
$$\Delta F_{Net-Ant}^{max} = \eta_{CF} \times \Delta F_{Net}^{max} \qquad (17)$$
Equation (17) assumes a linear dependence between cloud fraction and cloud radiative forcing
although, in nature, there could be three-dimensional radiation effects. In summer, $\eta_{CF}$ is

varying from 5% in Eastern Antarctica to 40% in Western Antarctica whilst, in winter, it is varying from 0% in Eastern Antarctica to 20% in Western Antarctica (Listowski et al., 2019). In December, if we consider $\eta_{CF}$ for SLW-containing cloud (that is to say both mixed-phase cloud and unglaciated SLW cloud consistent with our study), we find for a lower-level altitude cut-off of 0, 500 and 1000 m (Figure B1 in Listowski et al., 2019), a maximum SLWC radiative forcing $\Delta F_{Net-Ant}^{max}$ over Antarctica of about 12 W m$^{-2}$, 10 W m$^{-2}$ and 7 W m$^{-2}$, respectively. We now separate the Eastern elevated Antarctic Plateau from the Western Antarctica (Figure 5 in Listowski et al., 2019) for the 4 seasons. Over Eastern Antarctica, we find that $\Delta F_{Net-Ant}^{max} = 0.7$ to 7.0 W m$^{-2}$ in December-January-February (DJF) and 0 to 3.5 W m$^{-2}$ for the remaining seasons. Over Western Antarctica, the maximum radiative impact is much more intense because of higher temperatures and lower elevations compared to the Eastern Antarctic Plateau: $\Delta F_{Net-Ant}^{max} = 17.5$ to 40.0 W m$^{-2}$ in DJF (40 W m$^{-2}$ over the Antarctica Peninsula); 10.5 to 28.0 W m$^{-2}$ in March-April-May; 3.5 to 14.0 W m$^{-2}$ in June-July-August; and 7.0 to 17.5 W m$^{-2}$ in September-October-November. To summarize, the maximum SLWC radiative forcing over Western Antarctica (0 to 40 W m$^{-2}$) is estimated to 3 to 5 times larger compared to the one over the Eastern Antarctic Plateau (0 to 7 W m$^{-2}$), maximizing during the summer season.

**6. Conclusions**

Combining the observations of temperature, water vapour and liquid water path from a ground-based microwave radiometer, backscattering and depolarization from a ground-based LIDAR, screen-level air temperature and surface radiations at long and short wavelengths, our analysis has been able to evaluate the presence of supercooled liquid water clouds over the Dome C station in summer. Focusing on the month of December in 2018-2021, we established that in SLWCs temperature logarithmically increases from -36.0°C to -16.0°C when LWP increases from 1.0 to 14.0 g m$^{-2}$. We have also evaluated that SLWCs have a positive cloud

radiative forcing, which logarithmically increases from 0.0 to 70.0 W m$^{-2}$ when LWP increases
from 1.2 to 3.5 g m$^{-2}$. Our study clearly shows that SLWCs have a positive impact on $\Delta F_{LW}^{Down}$
increasing from 0 to 90 W m$^{-2}$ for LWP ranging from 1.0 to 3.5 g m$^{-2}$, a negligible impact ($\pm 5$
W m$^{-2}$) on $\Delta F_{LW}^{Up}$ for LWP ranging from 0 to 6.5 g m$^{-2}$, and a negative (but quite offsetting)
impact on each of the two terms $\Delta F_{SW}^{Down}$ and $\Delta F_{SW}^{Up}$ which decrease from 0 to -130 and -110
W m$^{-2}$, respectively for LWP ranging from 1.5 to 4.0 g m$^{-2}$. This means that the SLWC radiative
forcing is mainly driven by the downward surface irradiance since the attenuation of shortwave
incoming irradiance is almost compensated for by the upward shortwave irradiance because of
high values of surface albedo.

593        Finally, extrapolating our results of the SLWC radiative forcing from the Dome C station

to the Antarctic continent shows that the maximum SLWC radiative forcing is not greater than
7.0 W m$^{-2}$ over the Eastern Antarctic Plateau but 2 to 3 times larger (up to 40 W m$^{-2}$) over
Western Antarctica, maximizing over in summer season and over the Antarctic Peninsula. This
stresses the importance of accurately modelling SLWCs when calculating the Earth energy
budget to adequately forecast the Earth climate evolution, especially since the climate is rapidly
changing in Antarctica, as illustrated by the surface temperature record of -12°C recently
observed in March 2022 at the Concordia station and largely publicized worldwide (see e.g.
https://www.9news.com.au/world/antarctica-heatwave-extreme-warm-weather-recorded-
concordia-research-station/3364dd91-2051-4df5-8cfc-5f2819058604).

**Data availability**
HAMSTRAD data are available at http://www.cnrm.meteo.fr/spip.php?article961&lang=en
(last access: 27 November 2023). The tropospheric depolarization LIDAR data are reachable
at http://lidarmax.altervista.org/lidar/home.php (last access: 27 November 2023). Radiosondes
are available at http://www.climantartide.it (last access: 27 November 2023). Screen-level air
temperature from AWS can be obtained from the ftp server
(https://amrc.ssec.wisc.edu/data/archiveaws.html) (last access: 27 November 2023). BSRN
data can be obtained from the ftp server (https://bsrn.awi.de/data/data-retrieval-via-ftp/) (last
access: 27 November 2023).

**Author contribution**
PR, MDG, and AL provided the observational data. PR developed the methodology. All the
co-authors participated in the data analysis and in the data interpretation. PR prepared the
manuscript with contributions from all co-authors.

**Competing interests**
The authors declare that they have no conflict of interest.

**Acknowledgments**
The present research project Water Budget over Dome C (H2O-DC) has been approved by
the Year of Polar Prediction (YOPP) international committee. The HAMSTRAD programme
(910) was supported by the French Polar Institute, Institut polaire français Paul-Emile Victor
(IPEV), the Institut National des Sciences de l'Univers (INSU)/Centre National de la Recherche
Scientifique (CNRS), Météo-France and the Centre National d'Etudes Spatiales (CNES). The
permanently manned Concordia station is jointly operated by IPEV and the Italian Programma
Nazionale Ricerche in Antartide (PNRA). The tropospheric LIDAR operates at Dome C from
2008 within the framework of several Italian national (PNRA) projects. We would like to thank
all the winterover personnel who worked at Dome C on the different projects: HAMSTRAD,
aerosol LIDAR and BSRN. We would like to thank the three anonymous reviewers for their
beneficial comments.

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

# Tables

**Table 1.** Cloud-free periods in December 2018-2021 detected from the LIDAR depolarization observations at Concordia. Time is in UTC. MM-NN means from MM (included) hour UTC to NN (excluded) hour UTC. "X" means no cloud-free period during that day. "ND" means no LIDAR data available. Bold cases mean that cloud-free irradiance calculations are impossible due to lack of some data (LIDAR, HAMSTRAD, BSRN or AWS).

| Days | 2018 | 2019 | 2020 | 2021 |
|------|------|------|------|------|
| 01 | 0-24 | **9-18** | **ND** | **9-16** |
| 02 | 0-21 | **13-17** | **ND** | **7-8** |
| 03 | 0-24 | **6-16** | **ND** | **6-24** |
| 04 | X | **11-16** | **ND** | **0-24** |
| 05 | X | 6-16 | **3-16** | **12-19** |
| 06 | 3-6 | 0-13 | 9-13 | **2-12** |
| 07 | **1-16** | X | X | **0-24** |
| 08 | 3-15 | X | 1-2 | **0-10** |
| 09 | **2-16** | X | **4-14** | **10-17** |
| 10 | 0-3 | X | X | **ND** |
| 11 | X | 4-17 | 0-1 | **ND** |
| 12 | X | X | 20-22 | **ND** |
| 13 | 11-13 | 10-14 | 0-12 | X |
| 14 | 22-24 | 17-18 | X | 5-12 & 17-20 |
| 15 | 4-8 | **22-23** | X | 3-6 |
| 16 | **15-18** | X | **6-8** | 11-24 |
| 17 | 18-19 | **ND** | X | 0-24 |
| 18 | 1-17 | **ND** | 16-17 | 0-3 |
| 19 | 0-24 | **ND** | 7-9 & 11-13 | 20-23 |
| 20 | **0-12** | **ND** | 20-22 | 16-19 |
| 21 | X | **ND** | 20-21 | X |
| 22 | 9-16 | **ND** | ND | 12-15 |
| 23 | 1-4 | **ND** | **14-20** | X |
| 24 | X | **ND** | **11-14** | 0-6 |
| 25 | X | **ND** | 9-15 | **20-24** |
| 26 | 12-18 | **ND** | 0-16 & 18-22 | 0-24 |
| 27 | **10-11** | **ND** | 0-2 | 0-4 |
| 28 | **0-6** | **ND** | 0-17 | **10-14** |
| 29 | **X** | **ND** | 0-18 | **X** |
| 30 | **X** | **ND** | **7-24** | **X** |
| 31 | **10-12** | **ND** | 0-18 | **X** |


**Table 2.** Gaussian functions fitted to the $N(x)$ function for $x = T$ (°C) or $\Delta F$ (W m$^{-2}$). Units of
$a_1$, $a_2$, $a_3$, and $c_0$ are in count number for $T$ and $\Delta F$; units of $\mu_1$, $\mu_2$, $\mu_3$, $\sigma_1$, $\sigma_2$, and $\sigma_3$ are in
°C for $T$ and in W m$^{-2}$ for $\Delta F$.

| $x$ | $a_1$ | $\mu_1$ | $\sigma_1$ | $a_2$ | $\mu_2$ | $\sigma_2$ | $a_3$ | $\mu_3$ | $\sigma_3$ | $c_0$ |
|---|---|---|---|---|---|---|---|---|---|---|
| $T$ | 15.0 10$^3$ | -31.5 | 1.45 | 5.0 10$^3$ | -28.0 | 1.65 | 0.5 10$^3$ | -19.0 | 2.5 | -9.1 10$^{-6}$ |
| $\Delta F_{net}$ | 371.7 | 10.0 | 11.5 | 74.6 | 37.6 | 21.1 | 220.8 | 57.5 | 14.1 | -10.2 |
| $\Delta F_{LW}^{Down}$ | 415.5 | 10.0 | 10.4 | 189.5 | 53.7 | 24.2 | 227.1 | 82.9 | 7.0 | -18.5 |
| $\Delta F_{LW}^{Up}$ | - | - | - | - | - | - | - | - | - | - |
| $\Delta F_{SW}^{Down}$ | 190.5 | -10.1 | 17.2 | 113.0 | -80.0 | 54.6 | - | - | - | -1.9 |
| $\Delta F_{SW}^{Up}$ | 282.4 | -10.1 | 12.8 | 133.8 | -75.0 | 41.8 | - | - | - | 8.3 |



**Table 3.** Coefficients of the relations $f(LWP) = \alpha + \beta\ln(LWP)$ for the temperature $T$ or
cloud radiative forcing ($\Delta F_{net}$) and the individual components ($\Delta F_{LW}^{Down}$, $\Delta F_{LW}^{Up}$, $\Delta F_{SW}^{Down}$ and
$\Delta F_{SW}^{Up}$). Units of $T$ and $\Delta F$, as well as of their corresponding "$\alpha$" values are in °C and W m$^{-2}$,
respectively; units of $\beta$ are in °C g$^{-1}$ m$^2$ for $T$ and in W g$^{-1}$ for $\Delta F$; units of LWP are in g m$^{-2}$.
The last column shows the range of LWP values for which the relation is valid. $\alpha \pm \delta\alpha$
corresponds to the range of $\alpha$ values where the relationship is valid.

| $f(LWP)$ | $\alpha \pm \delta\alpha$ | $\beta$ | Valid range for $T$ or $\Delta F$ | Valid range for LWP |
|:---:|:---:|:---:|:---:|:---:|
| $T$ | -33.8 $\pm$ 1.5 | 6.5 | $[-36; -16]$ | $[1.0; 14.0]$ |
| $\Delta F_{net}$ | -18.0 $\pm$ 10.0 | 70.0 | $[0; 70]$ | $[1.2; 3.5]$ |
| $\Delta F_{LW}^{Down}$ | 5.0 $\pm$ 15.0 | 65.0 | $[0; 90]$ | $[1.0; 3.5]$ |
| $\Delta F_{LW}^{Up}$ | 0 $\pm$ 5.0 | 0.0 | $[-5; 5]$ | $[0.0; 6.5]$ |
| $\Delta F_{SW}^{Down}$ | 30.0 $\pm$ 30.0 | -130.0 | $[-130; 0]$ | $[1.5; 4.0]$ |
| $\Delta F_{SW}^{Up}$ | 30.0 $\pm$ 30.0 | -110.0 | $[-110; 00]$ | $[1.5; 4.0]$ |




# Figures

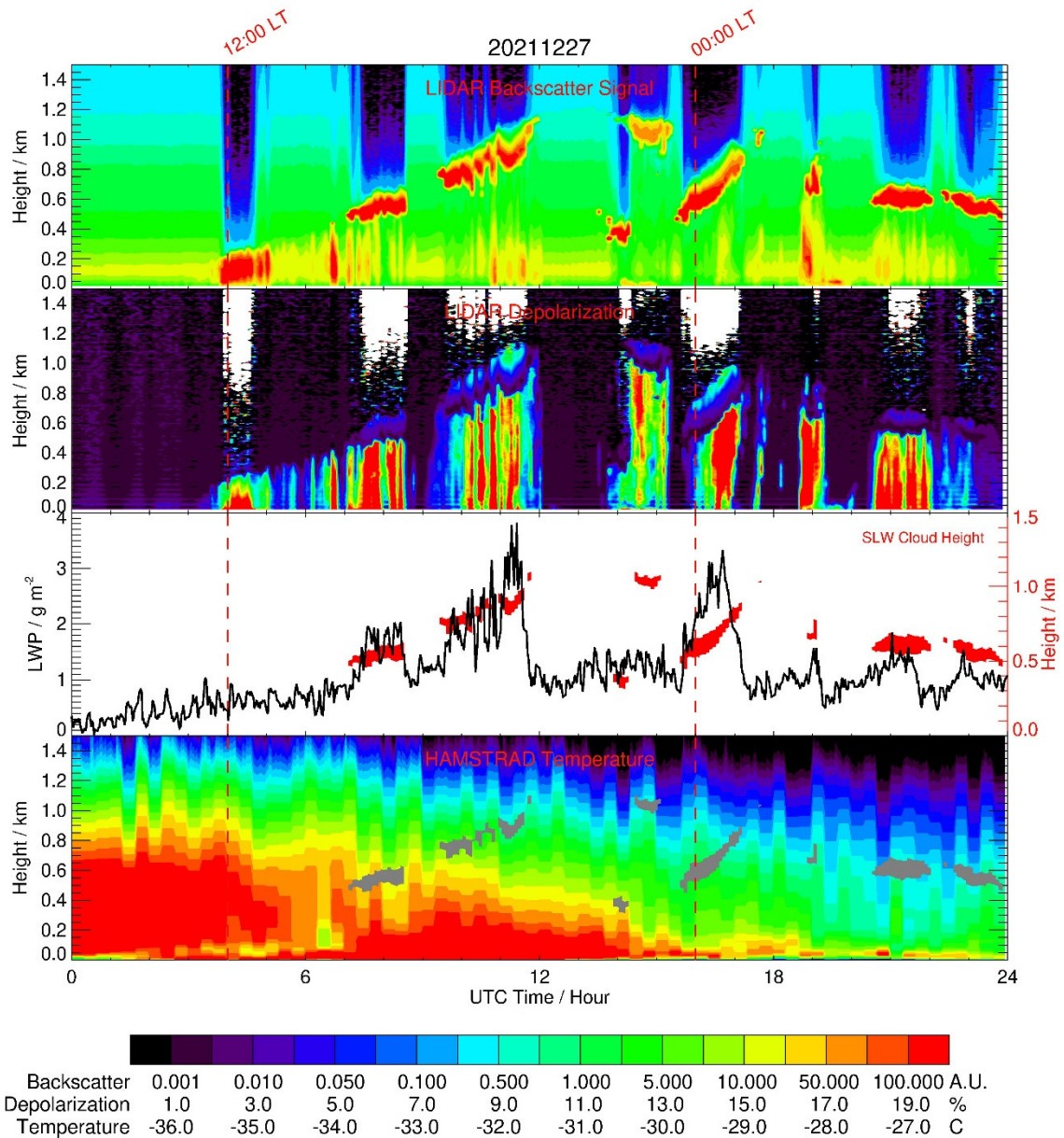

**Figure 1:** (From top to bottom): Time evolution (UTC, hour) of the LIDAR backscattering signal, the LIDAR depolarization signal, the HAMSTRAD LWP and the HAMSTRAD temperature profile measured on 27 December 2021. The time evolution of the SLW cloud (as diagnosed by a backscattering value > 60 A.U. and a depolarization value < 5%) is highlighted by the red and grey areas in the third and the forth panel from the top, respectively. The height above the ground is shown on the third panel from the top with the y-axis on the right. The 00:00 and 12:00 local times (LT) are highlighted by 2 vertical dashed lines.

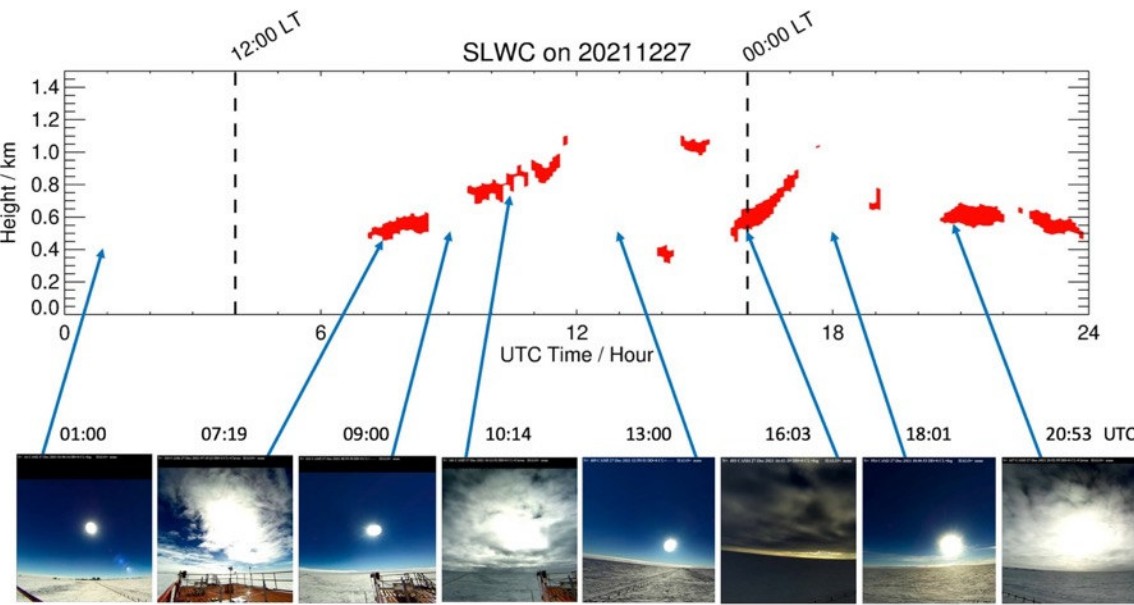

832

**Figure 2:** (Top) Time evolution (UTC, hour) of the SLWC (red areas) on 27 December 2021. (Bottom, from left to right) Snapshots from the HALO-CAM video camera taken on: 01:00 (no SLWC), 07:19 (SLWC), 09:00 (no SLWC), 10:14 (SLWC), 13:00 (no SLWC), 16:03 (SLWC), 18:01 (no SLWC) and 20:53 UTC (SLWC). The 00:00 and 12:00 local times (LT) are highlighted by 2 vertical dashed lines.

838

839

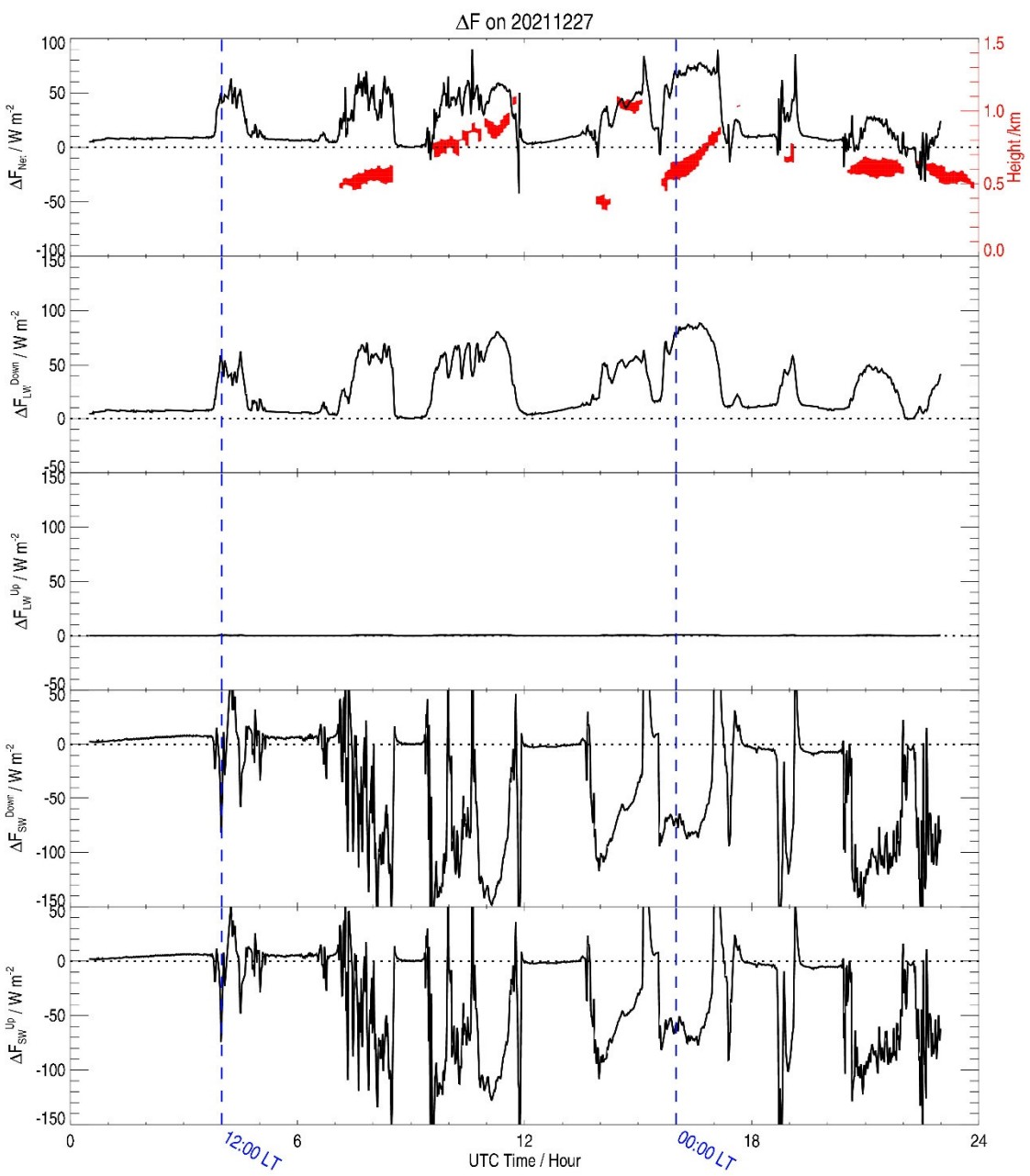

**Figure 3:** (from top to bottom) Time evolution (UTC, hour) of the cloud radiative forcing ($\Delta F_{net}$) (W m$^{-2}$) and its individual components: downward longwave ($\Delta F_{LW}^{Down}$), upward longwave ($\Delta F_{LW}^{Up}$), downward shortwave ($\Delta F_{SW}^{Down}$) and upward shortwave ($\Delta F_{SW}^{Up}$) calculated on 27 December 2021. The SLW cloud layer (if present) is highlighted by a red area in the uppermost panel, with the height on the y-axis shown on the right. The 00:00 and 12:00 local times (LT) are highlighted by 2 vertical blue dashed lines.

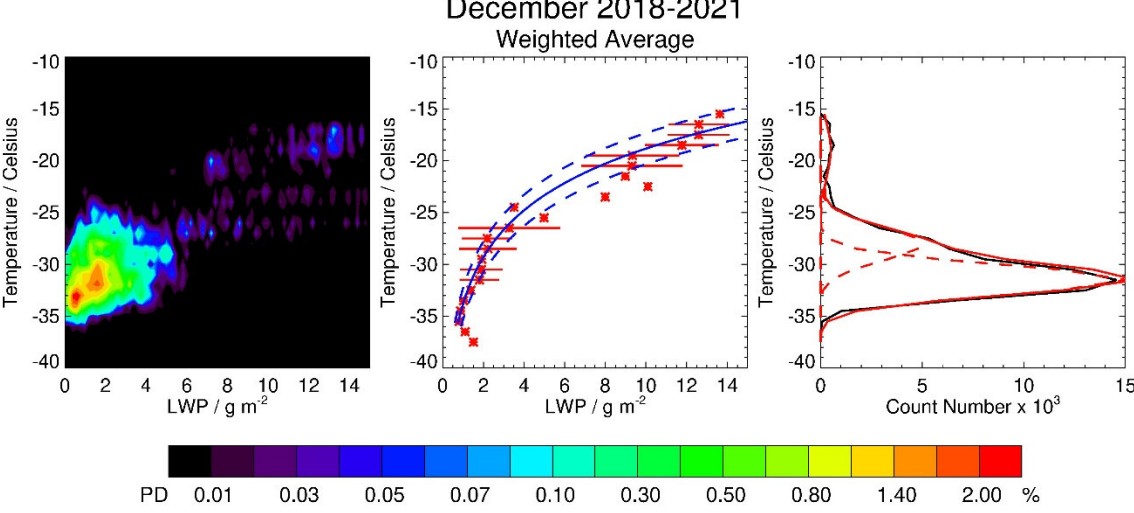

**Figure 4:** (Left) Probability Density (PD, %) of the temperature (°C) as a function of Liquid Water Path (LWP, g m$^{-2}$) in the SLWCs in December 2018-2021. The Probability Density is defined in the text. (Centre) Weighted-average LWP vs. temperature (red asterisks) with a fitted logarithmic function (blue solid) encompassing the significant points (within the two dashed blue lines). Horizontal bars represent 1-sigma variability in LWP per 1°C-wide bin. (Right) Temperature as a function of count number per 1°C-wide bin (black solid line) fitted with three Gaussian functions (red dashed curves). The sum of the three Gaussian functions is represented by a red solid line.

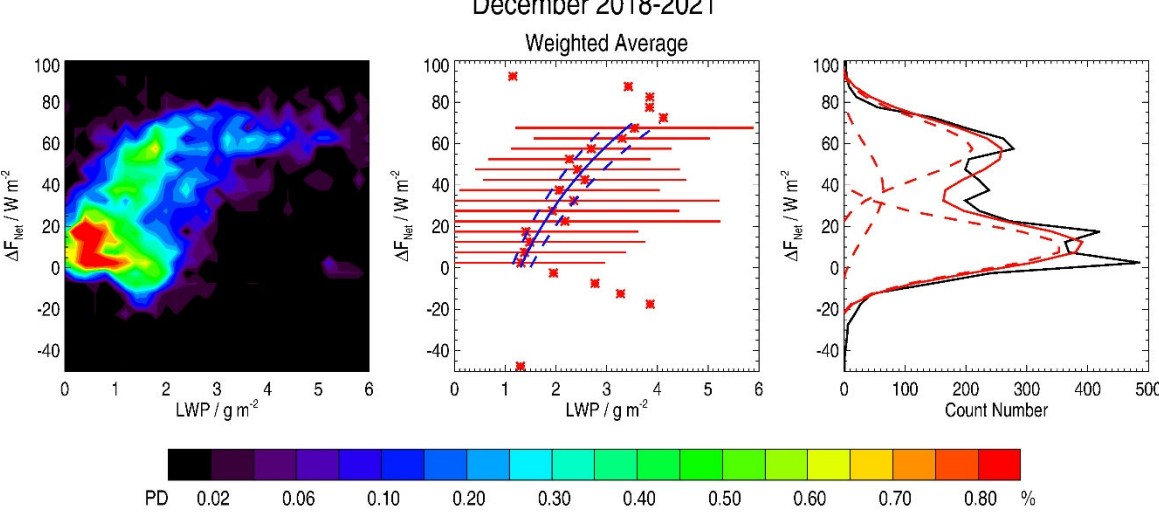

**Figure 5:** (Left) Probability Density (PD, %) of the cloud radiative forcing ($\Delta F_{net}$, W m$^{-2}$) as a function of Liquid Water Path (LWP, g m$^{-2}$) in the SLWCs in December 2018-2021. The Probability Density is defined in the text. (Centre) Weighted-average LWP vs. $\Delta F_{net}$ with a fitted logarithmic function (blue solid) encompassing the significant points (within the two dashed blue lines). Horizontal bars represent 1-sigma variability in LWP per 5 W m$^{-2}$-wide bin. (Right) $\Delta F_{net}$ as a function of count number per 5 W m$^{-2}$-wide bin (black solid line) fitted with three Gaussian functions (red dashed curves). The sum of the three Gaussian functions is represented by a red solid line.

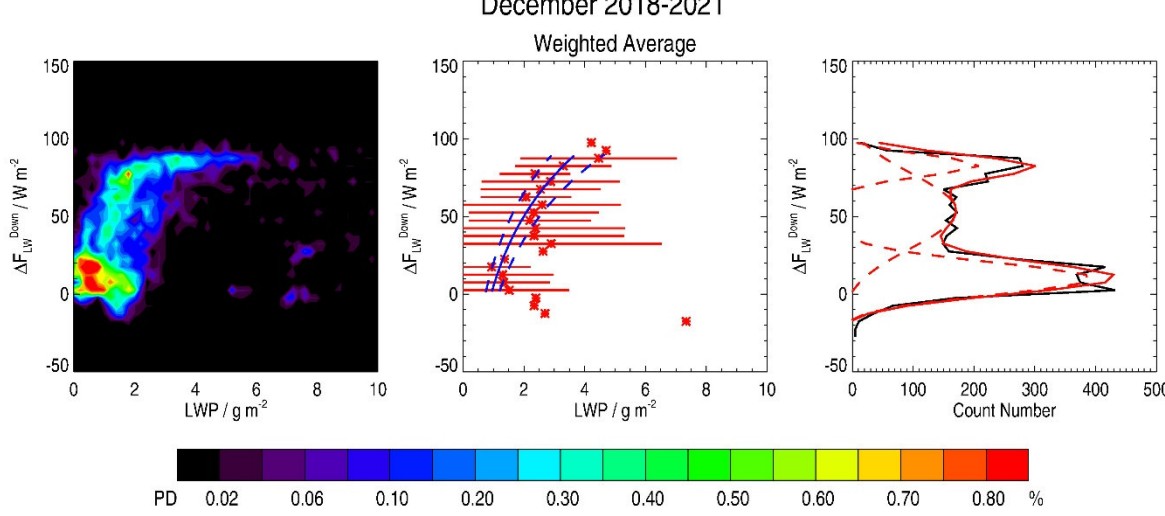


**Figure 6:** As in Figure 5 but for $\Delta F_{LW}^{Down}$.


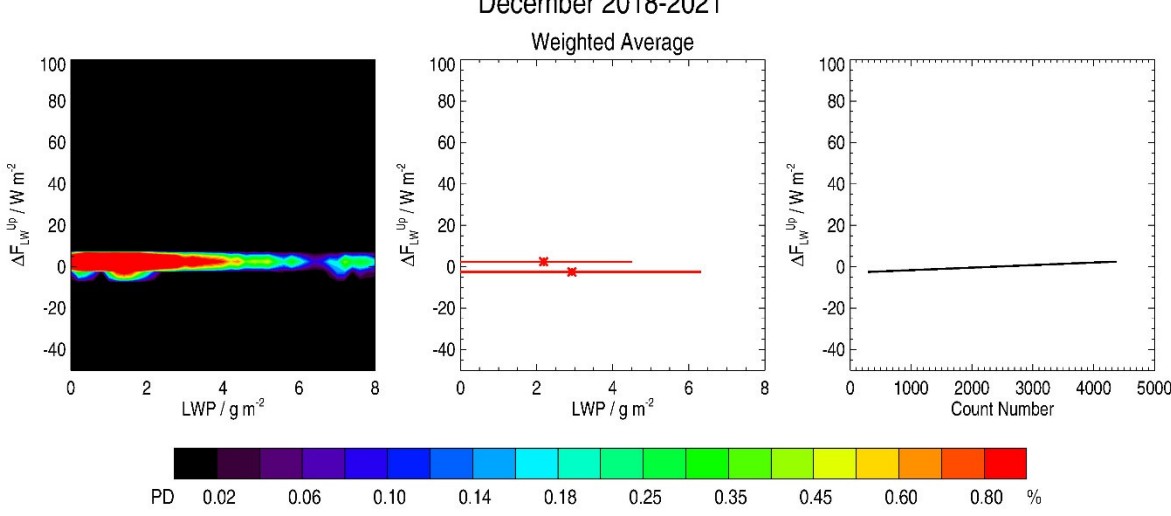


**Figure 7:** As in Figure 5 but for $\Delta F_{LW}^{Up}$.



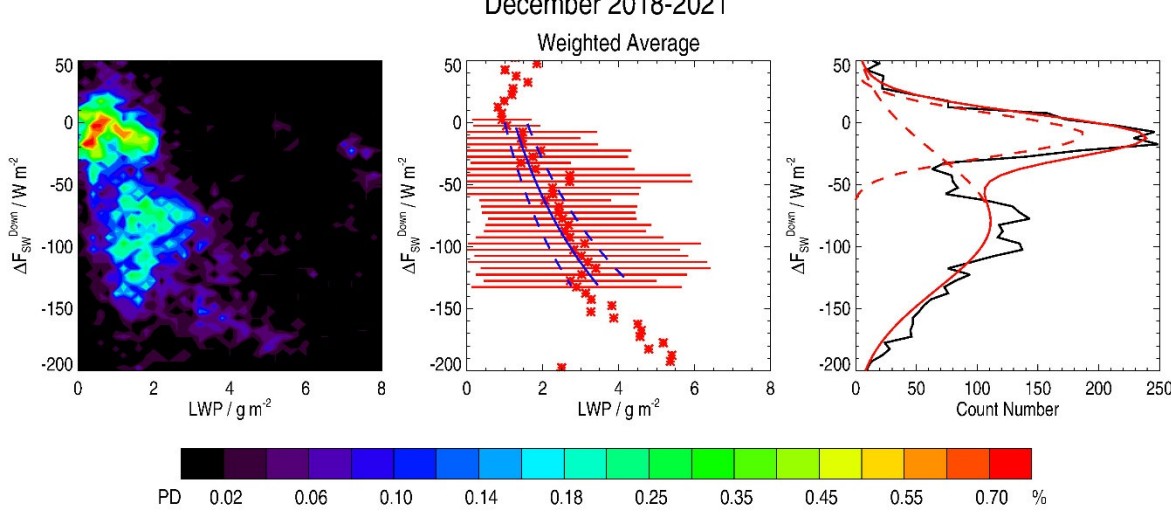


**Figure 8:** As in Figure 5 but for $\Delta F_{SW}^{Down}$.


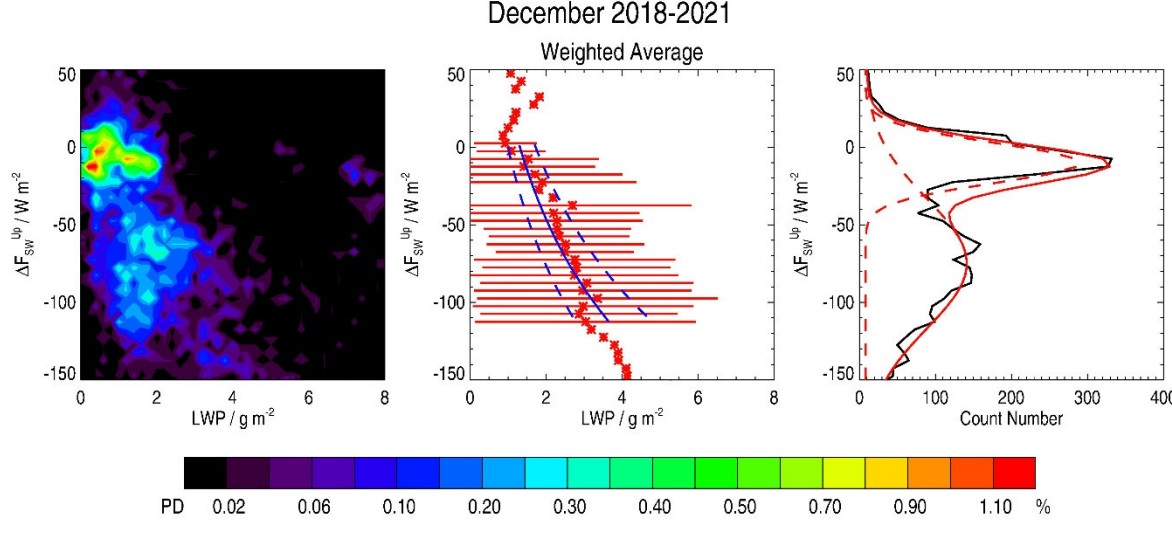


**Figure 9:** As in Figure 5 but for $\Delta F_{SW}^{Up}$.





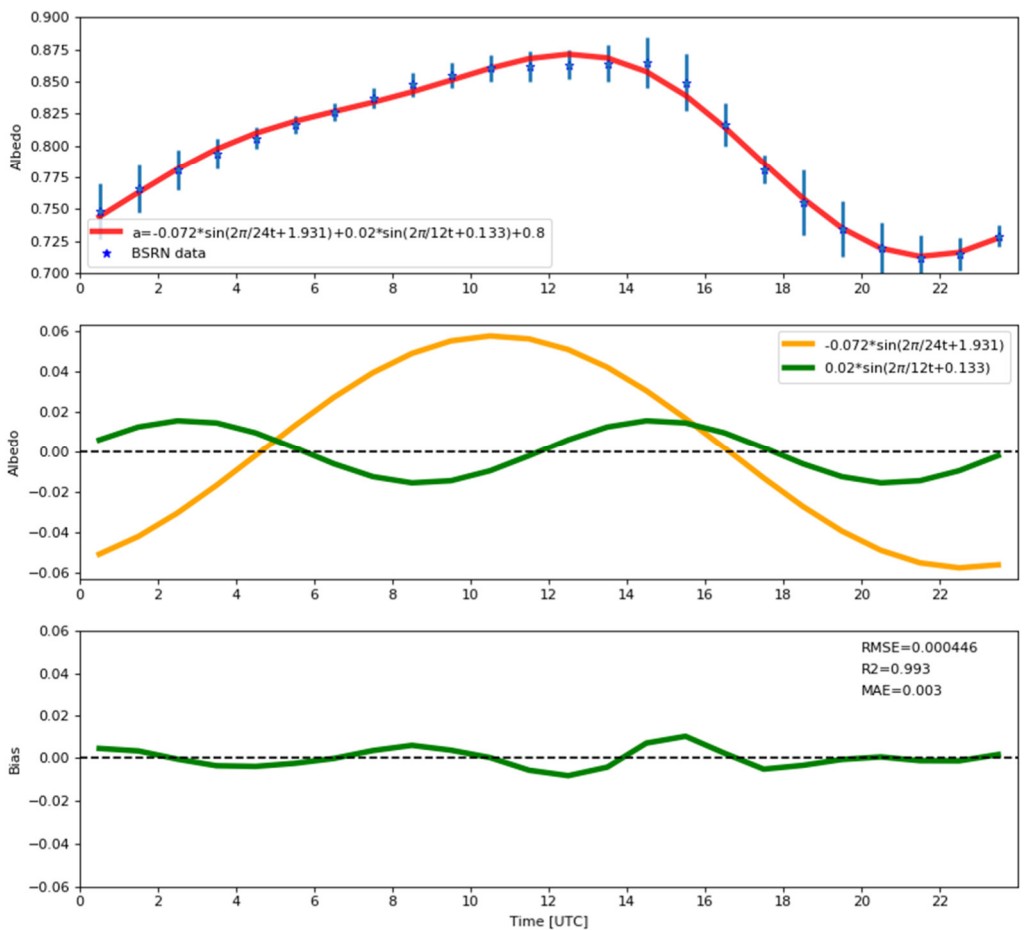


**Figure 10:** (Top) Hourly time evolution (UTC, hour) of the mean surface albedo observed by the BSRN instruments and the associated standard deviation (blue star and vertical bar, respectively) for the 5 cloud-free periods under consideration in our analysis together with the fitted trigonometric function based on 2 sine functions (red line). (Centre) The 2 sine functions fitting the hourly time evolution of the BSRN mean surface albedo. (Bottom) Hourly time evolution (UTC, hour) of the albedo residuals (BSRN-fit, green line) and corresponding values of associated Root Mean Square Error (RMSE), Coefficient of determination ($R^2$), and Mean Absolute Error (MAE).


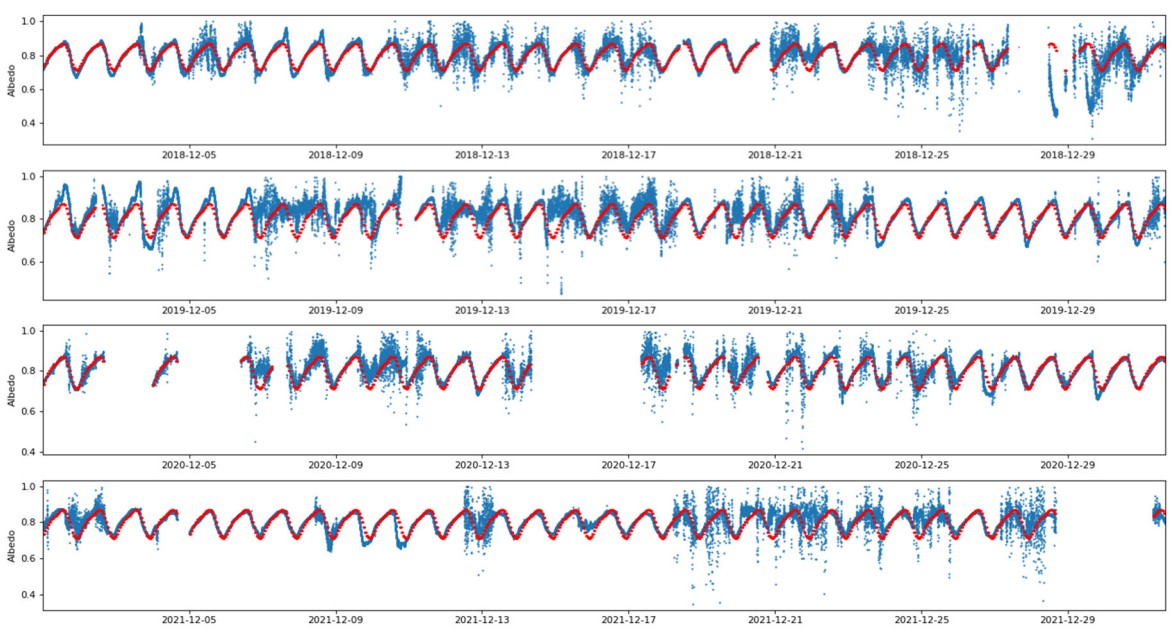


**Figure 11:** (from top to bottom) Hourly time evolution (UTC) of the surface albedo observed
by the BSRN instruments (blue), and using the fit based on 2 sine functions (red) for the whole
BSRN data set covering the month of December in: 2018, 2019, 2020 and 2021.