# Peer review of "Supercooled liquid water clouds observed over Dome C,"

_Atmospheric Chemistry and Physics, 2022_

## Author Response (AR1)

**Revision R01 Version 01, 1 December 2022**

**Manuscript Title:** *Supercooled liquid water clouds observed over Dome C, Antarctica: temperature sensitivity and surface radiation impact* **by Ricaud et al.**

**RESPONSES TO THE EDITOR**

→ Specific changes have been made in response to the reviewers' comments and are described below. The reviewers' comments are recalled in blue and changes in the revised version are highlighted in yellow. We have acknowledged the two anonymous reviewers. A sentence has been inserted in the Acknowledgements.

> We would like to thank the two anonymous reviewers for their beneficial comments.

**Reply to Anonymous Referee #1**

$\rightarrow$ Specific changes have been made in response to the reviewer's comments and are described below. The reviewer's comments are recalled in blue and changes in the revised version are highlighted in yellow.

**Summary**

The authors used ground-based remote sensing observations collected during the summertime over the course of 4 years at a continental site in Antarctica to explore the change in net surface fluxes caused by the presence of supercooled liquid-phase clouds. The authors find a strong relation between condensate amount (i.e., liquid water path, LWP) to ambient temperature. By using a clear-sky reference (i.e., selected days that were cloud-free), the authors extract changes in the surface radiative fluxes as a function of LWP and predict an Antarctic-wide change in the surface net radiative budget. The article is well written and contains interesting analysis. As listed below, there are several major concerns that make it difficult to support the authors' conclusions. Since the article touches on a highly relevant topic, I recommend a resubmission of this paper after resolving all major concerns.

$\rightarrow$ Thank you for your positive remarks in order to improve the quality of the manuscript. All your comments have been taken into account. An in-depth study has been performed to evaluate the choice of the clear-sky reference.

**Major concerns**

My main concern is the choice of clear-sky reference. Selecting clear days as reference has two major flaws: (1) the profiles of temperature, humidity, and aerosol properties may change drastically enough to introduce a bias when assessing clear-cloudy changes in surface radiation budget components; (2) this method deviates from traditional model-based assessments that simply perform radiative transfer calculations with and without cloud condensate on the same vertical profiles. A perfect clear-sky estimate would produce zero change in net surface radiative fluxes where clouds are absent. However, looking at Fig. 5 (top, e.g., 5-6 and 18-19 UTC) the change in non-zero. Similarly, relations shown in Fig. 7-9 (middle) show non-zero change in surface budget components when the LWP approaches zero. These examples hint at an underlying bias of the analysis. In their revision, the authors need to repeat their analysis using a typical broadband radiative transfer code (e.g., RRTMG) to verify the change in net radiative fluxes.

$\rightarrow$ This point is also a concern of the other reviewer, so we make a common response. We have performed, using several data sets, an in-depth study to evaluate the surface radiation components in clear-sky conditions.

1) Supplementary data sets

In order to evaluate the surface radiation in clear-sky conditions at Concordia, we have used, in complement to BSRN observations, and at the closest location to Concordia station, two different data sets of surface radiations from:

[revised manuscript text omitted]

***Figure R3.*** *Image of sastrugi on the ice surface.*

Figure R4 presents a satellite image showing the Concordia station. The sastrugi are clearly visible producing bright and dark straight lines with a 150°-330° orientation (wrt the N-S axis) (corresponding to solar azimuthal angles at ~13:00 and 01:00 LT). As a consequence, the albedo observed in BSRN is likely dependent of the sastrugi orientation, the sun elevation and the azimuthal angle.

[Figure]

***Figure R4.*** *Images taken from Google Earth showing the Concordia station on the right-hand side. The sastrugi effect producing bright and dark straight lines is clearly visible with one main orientation at ~330° orientation (~11:00 local time solar azimuth angle, with 0° orientation towards the North).*

If we suppose that the sastrugi effect impacts mostly SWU rather than SWD, and the albedo calculated from BSRN observations is the "truth", we can calculate a modified SWU* (including the sastrugi effect) for the ERA5 and CERES as:

SWU(ERA5)*=SWD(ERA5) x albedo(BSRN)
SWU(CERES)*=SWD(CERES) x albedo(BSRN)

Then we calculate the modified Net SR* (including the sastrugi effect) considering SWU* for ERA5 and CERES. As an example, we present Figure R5, similar to Figure R1, in which we added the albedo, the SWU* and Net SRs* (including the sastrugi effect) for CERES and ERA5 (solid lines). We observe that the Net SR* for ERA5 and CERES now coincides with the BSRN Net SR to within 5 W m$^{-2}$, compared to differences up to 50 W m$^{-2}$ found when the sastrugi effect was not taken into account.

[Figure]

***Figure R5.*** *Same as Figure R1 with the albedo inserted in the lowermost panel. Net SR, SWU SR, and albedo including the sastrugi effect for ERA5 (red solid line) and CERES (green solid line) have also been added in the Figures.*

We have fitted the BSRN albedo averaged over the 5 reference days with the sum of 2 sine functions, imposing periods of 24 and 12 hours. Figure R6 shows the BSRN albedo averaged over the five clear-sky days, the fitted trigonometric function and the residuals between the

averaged albedo and the fitted function. We can state that the sastrugi effect on the observed clear-sky albedo at Concordia is successfully fitted by 2 sine functions of 24h and 12h periods to within 0.003 mean absolute error, with a coefficient of determination $R^2$ equal to 0.993 and a root mean square error of 0.0004.

[Figure]

***Figure R6.*** *Top: Hourly time evolution (UTC, hour) of the mean surface albedo (blue stars) with the associated standard deviation (vertical bar) calculated from the BSRN data over the 5 clear-sky days together with the fitted trigonometric function made of two sine functions (red). Center: The two sine functions fitting the BSRN mean surface albedo. Bottom: bias of the fit curve (BSRN-fit) and associated root mean square error (RMSE), coefficient of determination ($R^2$), and mean absolute error (MAE).*

Moreover, we have considered all the BSRN observations in Decembers 2018, 2019, 2020 and 2021 to calculate the albedo (Figure R7), and we have superimposed the fitted trigonometric function as described in Figure R6. The presence of clouds is well highlighted by observations that depart from the fitted function whilst, during periods of clear-sky conditions, BSRN albedos coincide well with the fitted function.

[Figure]

***Figure R7.*** *Hourly time evolution (UTC, hour) of the surface albedo observed by the BSRN instruments (blue), and the two-sine fit (red) for the whole BSRN data set covering the month of December in 2018, 2019, 2020 and 2021.*

5) Conclusions

The study we have performed was extremely fruitful to evaluate the impact of the SLW clouds on the SR. The methodology requires reference clear-sky SR values that can be evaluated from: 1) models, 2) analyses and 3) observations. Our study has mainly shown that, at the Concordia station, sastrugi were present and strongly impacted the net SR via the surface albedo. This very local phenomenon cannot be taken into account by either the global-scale analyses (ERA5 and CERES), or standard radiative transfer models (e.g. RRTMG as suggested by the reviewers). As a consequence, the methodology we have developed based on field observations is likely the most powerful tool to estimate the Net SR in Concordia. It has some drawbacks, as for instance some biases for LWD and LWU between analyses and observations, but the LWD and LWU difference used to calculate the Net SR dramatically lessens the bias.

We have modified the revised version of the manuscript to explain the impact of the sastrugi effect on the SR by adding a detailed new section (see sub-section 5.2) and we have inserted a new paragraph in the abstract and in the conclusion.

Looking at Figure 5 (top) the change in net radiative fluxes is strongly affected by spikes in upward and downward shortwave radiative fluxes. While the reason for these spikes may simply be rooted in three-dimensional radiative effects, it is unclear how these spikes affect the analysis. The authors should justify their current approach or else find a way to filter for homogenous conditions or smooth the measured shortwave fluxes.

→ We are not sure to properly understand the question of the reviewer. These spikes on the SR signals are not artifacts. They come from the inhomogeneity of the three-dimensional radiation received by the instrument. They are absent when: 1) clear-sky conditions are encountered, and 2) clouds broadly cover the sky. Such spikes appear mainly 1) during scattered conditions and 2) when large cloud episodes appear or disappear. They are not restricted to the Concordia conditions but may appear at any latitude. For example, considering a similar instrumentation installed at the University of Toulouse, France, Figure R8 shows the time evolution of the SWD SR in clear-sky and cloudy conditions, together with the difference of the two (cloudy minus clear-sky signals). Negative as well as positive spikes appear in the difference when clouds are present likely due to incoming radiation reflected on cloud surface. Therefore, at Concordia, these spikes may also come from the inhomogeneity of the cloud distribution. As a conclusion, we do not consider that despiking is required because our approach is based on real field observations.

We mention this point in the revised manuscript (end of sub-section 3.1).

[Figure]

***Figure R8.*** *Time evolution of the shortwave downward surface radiation (W m$^{-2}$) observed in Toulouse (France) in clear-sky conditions on 13 June 2021 (top), in cloudy conditions on 26 June 2021 (middle) and difference between the two (bottom). Data time sampling is 10 sec.*

The article contains numerous instances of vague or non-scientific language that confuse when reading. For example, the authors refer to "liquid water concentration" (l. 79) which should probably be "liquid water content". Instead of listing a measurement of "radiation" (l. 92), the authors need to be more specific (i.e., "radiative flux" or "irradiance"). Measuring "aerosol and clouds" (l. 112) leaves a lot of room for speculation and needs to be specified. "Temperature" is a key metric and it is unclear whether the authors used potential temperature of liquid-water

potential temperature. The latter one is a conserved variable in a well-mixed, cloudy boundary layer and should be used. Using the former one (as maybe done here) would introduce a systematic error into the analysis that affect a key finding (i.e., a LWP-temperature relationship). The authors need to clarify which temperature was used and also over which altitudes the temperature was averaged (I'm guessing it is the cloud layer?).

→ We carefully checked the manuscript to avoid any confusion in terminology. We clarify below some points highlighted by the reviewer:

- We changed "liquid water concentration" into "liquid water content" as suggested.

- Our use of the term "radiation" is consistent with the terminology presented page 256 of Stull (1988). We have inserted the following sentence and reference in the revised manuscript.

  Hereafter, we will use either the term "radiative flux" or "radiation", the latter consistent with the terminology presented page 256 by Stull (1988).

  Stull, R. B.: An introduction to boundary layer meteorology. Kluwer Academic Publisher, 1988.

- We have used the regular temperature within the SLW cloud. We clarified this in the revised paper.

- (l. 112): we changed the incriminated text for "cloud characteristics" and we detail in the following paragraphs (see sub-section 2.1) of the manuscript which parameters are actually retrieved by the Lidar instrument.

Looking at Figure 1, there appears to be frozen precipitation below some clouds (e.g., 7-8 and 18:30 – 19:30 UTC) as indicated by relatively great backscatter and depolarization ration. It is unclear how (any) precipitation affects the HAMSTRAD liquid water content (or path) retrieval; the authors need to explain this. Furthermore, frozen precipitation hints at frozen particle inside clouds; the authors should discuss this issue and ideally refine their identification of supercooled clouds to ensure ice-free conditions.

→ Microwave observations at 60 and 183 GHz are not sensitive to ice crystals. This has already been discussed in Ricaud et al. (2018) when considering the study of diamond dust in Antarctica. As a consequence, precipitation detected by the Lidar does not affect retrievals of temperature, water vapour and liquid water. Nevertheless, during long episodes of large precipitation that can happen in winter periods, some amount of snow can accumulate on the shield protecting the HAMSTRAD radiometer. In summer time, a technician looks after the instrument and sweeps out the accumulated snow. But in winter, it is much more difficult for a technician to move outside of the base and unfortunately the sweeping may be performed few days after an intensive period of precipitation. As a consequence, snow that accumulates on the HAMSTRAD protective shield can perturb the 183-GHz signal, thus water vapour and LWP retrievals. But we recall that our present analysis is performed for the summer months of December 2018-2021 when precipitation is less intense and manpower very active to look after the instrument. We have added some sentences in the revised manuscript to clarify the issue of precipitation and impact on the LWP retrievals (see at the end of sub-section 2.2).

Another point is the ice-free conditions of the cloud we analyze. As mentioned in the text, we cannot state that the SLW clouds we study are 100% constituted of liquid water. Because of the methodology employed based on the depolarization ratio, the SLW clouds we observed are obviously made of liquid water. But there might be some periods when, associated with the SLW cloud, an ice layer is also detected above the SLW layer. We agree that this solid water layer may, at some stage, diffuse and/or precipitate within the SLW layer. The cloud in its whole might be formed of an ice layer superposed to a SLW layer. Unfortunately, in our present study, we cannot go through this macroscopic process by only using remote-sensed observations. In the discussion section 5.5 "Other clouds", these points were already presented.

**Minor concerns**

ll.. 53-54 While hydrometeors are larger, does the total water path change (or does it stay constant)? Please report.

→ This point has been dealt just above. Furthermore, hydrometeors are found to be larger on the coast than over the continent so the impact of the hydrometeors on LWP retrievals is even lower at Concordia than on the coast.

ll. 55-60 This sentence is too long and lacks clarity. For example, what object is referred to in "two to three times lower…".

→ The incriminated sentence has been changed into the Introduction as:

> Based on the raDAR/liDAR-MASK (DARDAR) spaceborne products (Listowski et al., 2019), it has been found that clouds are mainly constituted of ice above the continent. The abundance of Supercooled Liquid Water (SLW, the water staying in liquid phase below 0°C) clouds depends on temperature and liquid/ice fraction. It decreases sharply poleward, and is two to three times lower over the Eastern Antarctic Plateau than over the Western Antarctic.

ll. 68-69 This sentence makes is sound like supercooled water emerges from heterogenous nucleation. Please rephrase.

→ We rewrote the sentence into the Introduction as:

> Liquid water in clouds may occur in supercooled form due to a relative lack of ice nuclei for temperature greater than -39°C and less than 0°C.

l. 72 Please find an appropriate reference (perhaps within Storevlmo and Tan, 2015?).

→ We have added the 3 historical references in the revised manuscript and modified the sentences in the Introduction accordingly.

> Very little SLW is then expected because the ice crystals that form in this temperature range will grow at the expense of liquid droplets (called the "Wegener-Bergeron-Findeisen" process; Wegener, 1911; Bergeron, 1928; Findeisen, 1938; Storelvmo and Tan, 2015).

```
Bergeron, T., 1928: Über die dreidimensional verknüpfende
     Wetteranalyse. – Geophys. Norv.

Findeisen, W., 1938: Kolloid-meteorologische Vorgänge bei
     Niederschlagsbildung.   Meteorol.  Z.  55,  121–133.
     (translated and edited by Volken, E., A.M. Giesche, S.
     Brönnimann.    –    Meteorol.   Z.   24   (2015),
     DOI:10.1127/metz/2015/0675).

Wegener, A. 1911. Thermodynamik der Atmosphäre. – Leipzig,
     Germany: Barth.
```

l. 92 Please change "radiation" to "radiative flux".

→ Done

l. 112 Please specify "aerosol and clouds". Which properties were retrieved?

→ To be more focused on cloud characteristics, which is the main topic of the paper, we prefer not to detail Lidar capabilities for observing aerosols. We thus modified the term "aerosol and clouds" that is too vague into "cloud characteristics". The Lidar cloud observations are detailed farther in the manuscript (see sub-section 2.1).

l. 149 Please add "radiative" before "fluxes".

→ Done

ll. 191-192 This sentence needs rephrasing to improve readability.

→ We have modified the sentence into (sub-section 3.1):

```
Since it is impossible to measure for the same day the SR
with and without cloud, we have in priority looked for clear-
sky days over the months of December in the 2018-2021 period.
```

ll. 266-268 I'm assuming liquid-water potential temperature was used (i.e., a conserved variable in a cloudy boundary layer). Also, could these warm events also be moist – in other words, are there perhaps warm-moist intrusions that explain this LWP-temperature relation?

→ No, as explained in our response to a comment above and in the revised manuscript, we have not used liquid-water potential temperature but regular temperature.

Episodes of warm-moist intrusions exist above Concordia originated from mid-latitudes (Ricaud et al., 2017 and 2020) and known as "atmospheric rivers" (Wille et al., 2022). Although they are infrequent, they may provide high values of temperature and LWP. We have inserted a new sentence and a new reference associated to this point (see sub-section 4.1).

```
Wille, J.D., Favier, V., Dufour, A., Gorodetskaya, I.V.,
Turner, J., Agosta, C. and Codron, F., 2019. West Antarctic
```

surface    melt    triggered    by    atmospheric    rivers.    *Nature Geoscience*, *12*(11), pp.911-916.

ll. 300-302 This sentence contains too many verbs. Perhaps remove "is".

→ We modified the incriminated sentence and moved the corresponding paragraph to sub-section 5.1.

ll. 382-385 Not sure what "This" refers to. Please rephrase.

→ We have rephrased the sentence in the revised sub-section 5.5:

As  a  consequence,  the  presence  of  mixed-phase  clouds  in addition  to  SLWCs  may  explain  the  negative  part  of  the  Net, LWD  and  LWU  ΔSR  ([-20;0]   W m$^{-2}$)  and  the  positive  part  of  the SWD  and  SWU  ΔSR  ([0;10]    W m$^{-2}$)  for  low  values  of  LWP ([0.8;1.6]   g m$^{-2}$).

ll. 444-445 The link doesn't work. Please check.

→ We changed the link to "9news" since the link to "LA Times" did not work all the times:

https://www.9news.com.au/world/antarctica-heatwave-extreme-warm-weather-recorded-concordia-research-station/3364dd91-2051-4df5-8cfc-5f2819058604

Section 5.1 – The authors start a line of thinking that seems unfinished. I suggest the authors either flesh this out and compute whether a change in temperature truly explains the change in LWP or else shift the focus of this subsection to discuss the influence if warm-moist intrusions (as suggested above) or other meteorological drivers.

→ We have moved this section as a note in the new 5.2 section "Reference Surface Radiation and sastrugi effect".

Section 5.5 – To make this rough estimate believable the authors should report whether LWPs in other Antarctic regions are comparable.

→ The horizontal distribution and the temporal evolution of LWP over Antarctica is presented in Lenaerts et al. (2017) based on 2007-2010 reanalyses, observations and climate models. Over Antarctica, LWP is on average less than 10 g m$^{-2}$, with slightly larger values in summer than in winter by 2-5 g m$^{-2}$. Over the Western Antarctica, LWPs are larger (20-40 g m$^{-2}$) than over the Eastern Antarctica (0-10 g m$^{-2}$). As a consequence, LWP observed at Concordia is consistent with values observed over the Eastern Plateau, with a factor 2-4 smaller than those observed over the Western continent. We have inserted a new paragraph and a new reference in the revised section 5.6.

Lenaerts,  J.T.,  Van  Tricht,  K.,  Lhermitte,  S.  and  L'Ecuyer, T.S.,  2017.  Polar  clouds  and  radiation  in  satellite observations,  reanalyses,  and  climate  models.  Geophysical Research Letters, 44(7), pp.3355-3364.

Figure 1 - Please improve the lower end of the color scale for depolarization ratio (i.e., near 5% that is used as threshold).

→ We have modified the Figure accordingly (see Figure R9 below).

[Figure]

Figure R9: (From top to bottom): Time evolution (UTC, hour) of the Lidar Backscattering Signal, the Lidar Depolarization Signal, the HAMSTRAD LWP and the HAMSTRAD temperature profile measured on 27 December 2021. The time evolution of the SLW cloud (as diagnosed by a backscattering signal > 60 A.U. and a depolarization signal < 5%) is highlighted by the red and grey areas in the third and the forth panel from the top, respectively. The height above the ground is shown on the third panel from the top with the y-axis on the right. The 00:00 and 12:00 local times (LT) are highlighted by 2 vertical dashed lines.

Figure 2 – The detailed images should be shown at higher quality to enable a visual assessment of the presence of ice-cloud features (e.g., halo). Also an image should be shown for the period 4-5 UTC that was brought up in the main text.

→ The images come from an automated camera and this is the highest quality we can have. The point is that we show only 2 sets of images: 1) cloud-free and 2) SLWCs. We do not show any ice cloud so it is not expected to visualize any halo.

Unfortunately, the camera stopped acquiring images between 03:47:16 and 05:35:11 UTC. So we cannot insert any other image for this period in Figure 2.

Figure 6 – It is unclear which temperature is shown (absolute, potential, or liquid-water potential temperature) and which layers were used for averaging (e.g., in-cloud layers only?).

→ This point is dealt above. We clarified the methodology employed. We show regular temperature. There is no averaging at all. The Lidar profiles are interpolated along the temperature vertical grid and then according to the temperature time sampling. As a consequence, for a given time and height, we have a depolarization ratio, a backscatter signal, a regular temperature and a (not height-dependent) LWP. The same method is used for SR. BSRN SRs are time interpolated to be coincident with the LWP values. As a consequence, for a given time, we have a set of BSRN SRs (Net, LWU, LWD, SWU and SWD) and an LWP. At a (time, height) point showing high backscatter signal and low depolarization, the associated parameters (regular temperature, LWP and SRs) are flagged as "SLW cloud". The statistic is thus done using all the SLW-flagged points without any averaging. The temperature corresponds to the in-cloud temperature.

The text has been modified accordingly (see sub-section 3.1)

Figure 6-9 – It is unclear why the authors decided to determine statistics (i.e., the bars) in horizontal direction. Isn't the question for example in Figure 7 "which change in net radiative budget corresponds to a certain LWP" which would call for a statistic per LWP (i.e., producing a vertical bar)?

→ We agree that it could have been interesting to make the analysis in the way suggested by the reviewer. However, we wanted to study the sensitivity of LWP for a given temperature and for a given radiation component (Net, LWD, LWU, SWD, SWU). We inserted a sentence in the revised manuscript (see sub-section 3.2).

> This study is focused on the evaluation of the LWP sensitivity for a given temperature and for a given radiation component (Net, LWD, LWU, SWD, SWU).

**Reply to Anonymous Referee #2**

→ Specific changes have been made in response to the reviewer's comments and are described below. The reviewer's comments are recalled in blue and changes in the revised version are highlighted in yellow.

Review of „Supercooled liquid water clouds observed over Dome C, Antarctica: temperature sensitivity and surface radiation impact" by Ricaud et al.

The authors investigate the radiative impact of SCLW clouds at Dome C. In general, the topic is very interesting and suitable for ACP and the paper is well written. However, I have some concerns regarding the methodology and recommend major revision.

→ Thank you for your positive remarks in order to increase the quality of the manuscript. All your comments have been taken into account. An in-depth study has been performed to evaluate the choice of the clear-sky reference.

Major comments:

Estimating clear sky surface radiation: Instead of averaging the observations of the complete day, the authors could fit a sinus function to the clear sky observations to account for the diurnal cycle. Likely, this would reduce the bias of up to 20 w/m2 that can be seen in Fig. 4, top panel. Also, when using averaged observations instead of a radiative transfer model to estimate clear sky radiation, the authors should quantify the uncertainties caused by varying temperature and aerosol profiles among the elected clear sky days. A radiative transfer model would be required for this which leaves the question why such a model is not used in the first place.

→ This point has been studied in detail and according to concurring comments from the Reviewer #1. We have performed, using several data sets, an in-depth study to evaluate the surface radiation components in clear-sky conditions. We therefore reproduce below our common response to the two Reviewers.

1) Supplementary data sets

In order to evaluate the surface radiation in clear-sky conditions at Concordia, we have used, in complement to BSRN observations, and at the closest location to Concordia station, two different data sets of surface radiations from:

a) the European Center for Medium-Range Weather Forecasts Reanalysis version 5 (**ERA5**). ERA5 is a climate reanalysis dataset, covering the period 1979 to present. ERA5 is being developed through the Copernicus Climate Change Service (C3S). Extracted data (https://cds.climate.copernicus.eu/cdsapp#!/dataset/reanalysis-era5-single-levels) used here are hourly at a regular horizontal grid of 0.25°x0.25° in clear-sky conditions: surface solar and thermal infrared, downward and net radiations. As explained on the ERA5 website, clear-sky radiations are computed for the same atmospheric conditions of temperature, humidity, ozone, trace gases and aerosol as the corresponding total-sky quantities (clouds included), but assuming that the clouds are not there.

b) the Clouds and the Earth's Radiant Energy System (**CERES**), containing SYN1deg (Hourly CERES and geostationary (GEO) TOA fluxes, MODIS/VIIRS and GEO cloud properties, MODIS/VIIRS aerosols, and Fu-Liou radiative transfer surface and in-atmospheric (profile) fluxes consistent with the CERES observed TOA fluxes, as explained on https://ceres.larc.nasa.gov/data/). Surface fluxes in SYN1deg are computed with cloud properties derived from MODIS and geostationary satellites (GEO), where each geostationary satellite instrument is calibrated against MODIS (Doelling et al. 2013; 2016) at 1°x1° horizontal resolution (https://ceres.larc.nasa.gov/data/). Aerosol and atmospheric data were included as inputs to calculate the radiation flux.

We have compared the CERES and ERA5 data with the BSRN hourly-averaged data on the 5 reference days (clear-sky conditions) for the Net, LWD, LWU, SWD and SWU SRs. Figure R1 shows these variables for the 26 December 2021. The LWD and LWU values show an overall consistency between ERA5 and CERES (of the order of ~10 W m$^{-2}$), while a systematic negative bias of ~20-40 W m$^{-2}$ is observed with respect to BSRN data. However, the net longwave radiation, i.e. the difference LWD – LWU for each data set, is reduced to around 5 W m$^{-2}$. The SWD and SWU signals from ERA5, CERES and BSRN show a similar diurnal variation with differences less than 50 W m$^{-2}$. When considering the Net SR, some obvious differences up to 50 W m$^{-2}$ can be seen between BSRN, ERA5 and CERES. Since the net longwave radiation is within 10 W m$^{-2}$ for the three data sets, the source of this difference therefore should come from either SWD or SWU radiation.

[Figure]

***Figure R1.*** *Hourly time evolution (UTC, hour) of the clear-sky surface radiations (SR, W m⁻²) observed by the BSRN instruments (blue asterisks), the CERES (green asterisks) and the ERA5 (red asterisks) data sets on 26 December 2021: (from top to bottom) Net SR, Longwave Downward (LWD SR), Longwave Upward (LWU SR), Shortwave Downward (SWD SR) and Shortwave Upward (SWU SR) surface radiations. The 00:00 and 12:00 local times (LT) are highlighted by 2 vertical dashed lines.*

3) Impact of the local albedo on shortwave radiation.
We have calculated, for BSRN, ERA5 and CERES data, the albedo defined as:

Albedo=SWU/SWD.

Figure R2 shows the diurnal evolution of the albedo on 26 December 2021 (clear-sky day). The CERES and ERA5 albedos do not show any significant diurnal variation with quite constant values of 0.74 and 0.83, respectively, whilst the observed BSRN albedo shows a clear diurnal

signal with a maximum of 0.85 from 10:00 to 14:00 UTC (from 18:00 to 22:00 LT) and a minimum of 0.70 from 19:00 to 23:00 UTC (from 03:00 to 07:00 LT). The large diurnal signal present in the observed albedo is likely the signature of the sastrugi effect that is obviously absent in the ERA5 and CERES data sets. The BSRN SWU sensor has a circular footprint. For a sensor installed at a height $h$ above the ground, 90% of the signal comes from an area at the surface closer than 3.1 $h$ (Kassianov et al., 2014). Since at Dome-C the instrument is installed at a height of 2-3 m, the albedo is thus determined by the surface elements in the immediate vicinity (a few meters) of the sensor.

Kassianov E, Barnard J, Flynn C, Riihimaki L, Michalsky J, Hodges G (2014) Areal-averaged spectral surface albedo from ground-based transmission data alone: toward an operational retrieval. Atmosphere 5:597–621. https://doi.org/10.3390/atmos503059)

[Figure]

***Figure R2.*** *Time evolution (UTC, hour) of the surface albedo observed by the BSRN sensors (blue), the CERES (green) and the ERA5 (red) data sets on 26 December 2021. The 00:00 and 12:00 local times (LT) are highlighted by 2 vertical dashed lines.*

4) Impact of sastrugi on the albedo.

Sastrugi (Figure R3) are features formed by erosion of snow by wind. They are found in polar regions, and in snowy, wind-swept areas of temperate regions, such as frozen lakes or mountain ridges. Sastrugi are distinguished by upwind-facing points, resembling anvils, which move downwind as the surface erodes.

[Figure]

***Figure R3.*** *Image of sastrugi on the ice surface.*

Figure R4 presents a satellite image showing the Concordia station. The sastrugi are clearly visible producing bright and dark straight lines with a 150°-330° orientation (wrt the N-S axis) (corresponding to solar azimuthal angles at ~13:00 and 01:00 LT). As a consequence, the albedo observed in BSRN is likely dependent of the sastrugi orientation, the sun elevation and the azimuthal angle.

[Figure]

***Figure R4.*** *Images taken from Google Earth showing the Concordia station on the right-hand side. The sastrugi effect producing bright and dark straight lines is clearly visible with one main orientation at ~330° orientation (~11:00 local time solar azimuth angle, with 0° orientation towards the North).*

If we suppose that the sastrugi effect impacts mostly SWU rather than SWD, and the albedo calculated from BSRN observations is the "truth", we can calculate a modified SWU* (including the sastrugi effect) for the ERA5 and CERES as:

SWU(ERA5)*=SWD(ERA5) x albedo(BSRN)
SWU(CERES)*=SWD(CERES) x albedo(BSRN)

Then we calculate the modified Net SR* (including the sastrugi effect) considering SWU* for ERA5 and CERES. As an example, we present Figure R5, similar to Figure R1, in which we added the albedo and the SWU* and Net SRs* (including the sastrugi effect) for CERES and ERA5 (solid lines). We observe that the Net SR* for ERA5 and CERES now coincides with the BSRN Net SR to within 5 W m$^{-2}$, compared to differences up to 50 W m$^{-2}$ found when the sastrugi effect was not taken into account.

[Figure]

***Figure R5.*** *Same as Figure R1 with the albedo inserted in the lowermost panel. Net SR, SWU SR, and albedo including the sastrugi effect for ERA5 (red solid line) and CERES (green solid line) have also been added in the Figures.*

We have fitted the BSRN albedo averaged over the 5 reference days with the sum of 2 sine functions, imposing periods of 24 and 12 hours. Figure R6 shows the BSRN albedo averaged over the five clear-sky days, the fitted trigonometric function and the residuals between the

averaged albedo and the fitted function. We can state that the sastrugi effect on the observed clear-sky albedo at Concordia is successfully fitted by 2 sine functions of 24h and 12h periods to within 0.003 mean absolute error, with a coefficient of determination $R^2$ equal to 0.993 and a root mean square error of 0.0004.

[Figure]

***Figure R6.*** *Top: Hourly time evolution (UTC, hour) of the mean surface albedo (blue stars) with the associated standard deviation (vertical bar) calculated from the BSRN data over the 5 clear-sky days together with the fitted trigonometric function made of two sine functions (red). Center: The two sine functions fitting the BSRN mean surface albedo. Bottom: bias of the fit curve (BSRN-fit) and associated root mean square error (RMSE), coefficient of determination ($R^2$), and mean absolute error (MAE).*

Moreover, we have considered all the BSRN observations in Decembers 2018, 2019, 2020 and 2021 to calculate the albedo (Figure R7), and we have superimposed the fitted trigonometric function as described in Figure R6. The presence of clouds is well highlighted by observations that depart from the fitted function whilst, during periods of clear-sky conditions, BSRN albedos coincide well with the fitted function.

[Figure]

**Figure R7.** *Hourly time evolution (UTC, hour) of the surface albedo observed by the BSRN instruments (blue), and the two-sine fit (red) for the whole BSRN data set covering the month of December in 2018, 2019, 2020 and 2021.*

5) Conclusions

The study we have performed was extremely fruitful to evaluate the impact of the SLW clouds on the SR. The methodology requires reference clear-sky SR values that can be evaluated from: 1) models, 2) analyses and 3) observations. Our study has mainly shown that, at the Concordia station, sastrugi were present and strongly impacted the net SR via the surface albedo. This very local phenomenon cannot be taken into account by either the global-scale analyses (ERA5 and CERES), or standard radiative transfer models (e.g. RRTMG as suggested by the reviewers). As a consequence, the methodology we have developed based on field observations is likely the most powerful tool to estimate the Net SR in Concordia. It has some drawbacks, as for instance some biases for LWD and LWU between analyses and observations, but the LWD and LWU difference used to calculate the Net SR dramatically lessens the bias.

We have modified the revised version of the manuscript to explain the impact of the sastrugi effect on the SR by adding a detailed new section (see sub-section 5.2) and we have inserted a new paragraph in the abstract and in the conclusion.

Measurement errors: What are the typical measurement errors of the instruments and how do they impact the results of the study? Also, the sometime negative radiation balance should lead to near-surface temperature inversions that are hard to measure with microwave radiometers because the obtained temperature profiles are typically very smooth. How do these inversions impact the results?

→ We kept in the revised manuscript a sub-section "Errors" (5.4) that was already in the submitted manuscript. It is dedicated to measurement errors associated to each instrument. We highlighted the fact that random errors cannot impact on the results presented here because they are divided by $N^{1/2}$ with N the number of observations. It is clear that systematic errors are not

altered by the statistics but these errors are extremely difficult to characterize and may provide an explanation of the small biases found. This point has already been discussed.

Regarding temperature, we are not quite sure about the meaning of the question. We select in-cloud temperature to perform our analyses. Since SLW clouds are generally present in an altitude range from 400 to 800 m in December, the near-surface temperature profiles (and more specifically temperature inversion) would have a negligible effect on the observed parameters in the cloud layer.

Minor comments:

General: When discussing temperature, I recommend to be more specific. Is it surface or cloud temperature? Is it potential or regular temperature?

→ This point is presented in detail in the replies to the comments from Reviewer #1 together with the methodology employed. We use in-cloud regular temperature.

L94: Readability would improve if proper notation was used.

→ We have modified the notations accordingly (see in the Introduction).

L236: What is the physical explanation why multiple normal distributions are required?

→ It is rather difficult to give a simple explanation. Considering the SR vs. LWP relationship, it seems that we have systematically one of the Gaussian distributions centred around 0 W m$^{-2}$, reflecting the non-impacting part of SLWCs on SR components. Considering the temperature vs. LWP relationship, the 2 main Gaussian distributions are centred around -28°C and -30°C, corresponding to temperatures usually encountered in Concordia whilst the third one, far much less intense, is centered around -18°C, probably the signature of very unusual events occurring in Concordia as the warm-moist events associated with atmospheric rivers. We have inserted two paragraphs in the revised manuscript (see sub-section 4.2).

L254: I think the plots show the probability density, but not the probability density function which would be an analytical formula to describe the probability density.

→ Right. We modified the term "Probability Density Function" into "Probability Density".

L261: *joint* Gamma distributions ?

→ We clarified this issue by rewriting the sentence as (see sub-section 4.1):

> We have performed a weighted average of the LWPs within each temperature bin (Figure 6 centre). Then, we have fitted 3 Gaussian distributions to the count numbers as a function of temperature (Figure 6 right). If we now only consider temperature bins within one-sigma of the centre of the Gaussian distributions, we can fit the following logarithmic relation of the temperature $\theta$ as a function of LWP within the SLWC (Figure 6 centre):

Similarly, we have modified the sentence related to the Gaussian distribution of SR vs LWP as (see sub-section 4.2):

> The central panel shows, for the same parameters, the corresponding weighted average LWP within 5 W m$^{-2}$-wide bins of radiation anomaly whereas the right panel shows the corresponding count number within 5 W m$^{-2}$-wide bins fitted by 2 Gaussian distributions.

L265, L281: add unit to +/- 1.5, +/- 10

→ Done: ($\pm 1.5$ °C) and ($\pm 10.0$ W m$^{-2}$)

L267: liquid water *content*

→ Done

L354: What variables would be how impacted by ice contaminated clouds?

→ This point has been dealt in detail while considering a concurring comment from the Reviewer #1. Here is our common response:
→ Microwave observations at 60 and 183 GHz are not sensitive to ice crystals. This has already been discussed in Ricaud et al. (2018) when considering the study of diamond dust in Antarctica. As a consequence, precipitation detected by the Lidar does not affect retrievals of temperature, water vapour and liquid water. Nevertheless, during long episodes of large precipitation that can happen in winter periods, some amount of snow can accumulate on the shield protecting the HAMSTRAD radiometer. In summer time, a technician looks after the instrument and sweeps out the accumulated snow. But in winter, it is much more difficult for a technician to move outside of the base and unfortunately the sweeping may be performed few days after an intensive period of precipitation. As a consequence, snow that accumulates on the HAMSTRAD protective shield can perturb the 183-GHz signal, thus water vapour and LWP retrievals. But we recall that our present analysis is performed for the summer months of December 2018-2021 when precipitation is less intense and manpower very active to look after the instrument. We have added some sentences in the revised manuscript to clarify the issue of precipitation and impact on the LWP retrievals (see sub-section 2.2).

L409ff: Would the mean Net SR be more interesting than the maximum?

→ Since we are interested in the maximum possible impact of SLWC on Net SR, it sounds more logical to study the maximum of Net SR. Moreover, if the impact had appeared to be negligible when considering the maximum Net SR, we could have concluded that the impact of SLWC on the Net SR was limited.

L410: Please provide the mean cloud fraction value.

→ We have inserted a new sentence (see sub-section 5.6):

> In summer, $\eta_{CF}$ is varying from 5% in Eastern Antarctica to 40% in Western Antarctica whilst, in winter, it is varying from 0% in Eastern Antarctica to 20% in Western Antarctica (Listowski et al., 2019).

---

## Referee Report (RR1)

Referee comment on "Supercooled liquid water clouds observed over Dome C, Antarctica: temperature sensitivity and surface radiation impact" by Ricaud et al.

The study investigates the relation between temperature and liquid water path in the presence of supercooled liquid water clouds, and analyzed the cloud radiative effect in terms of net surface radiation.

In general, the authors have attempted to thoroughly revise their manuscript according to the reviewers' comments. The relationship between temperature and LWP is well described. However, I have still some concerns regarding the methodology used to estimate the cloud radiative effect. Overall, I cannot recommend publication without a major revision.

Major Comments

1. In the revised version, the authors included ERA5 and CERES data to evaluate the surface radiation in cloudless conditions. I don't think this significantly supports the study for several reasons. First, a potential effect of surface roughness (sastrugi) on surface albedo could be directly estimated from the pyranometer measurements. Second, it is not clear to me, for what reason ERA5 and CERES data were introduced here. It is known, that these data sets are not very reliable in representing the surface albedo, as also shown by the authors. Both data sets were fudged to the measurements by using the measured surface albedo for scaling the upward irradiance. What is to be done with the fitted results? How does it help to estimate the radiative effect of clouds based on the ERA5/CERES data, since they are still based on the near-time measured albedo? How do they prove the stability of conditions?

2. A more feasible approach was already suggested by the previous reviewers. I strongly encourage the authors to use radiative transfer modeling to estimate the cloudless reference surface radiation. Since the surface albedo are available directly from the measurements in cloudless conditions, even the diurnal pattern can be considered. Note, that snow/white ice albedo is also dependent on the solar zenith angle (e.g., Gardner and Sharp, 2010 - https://doi.org/10.1029/2009JF001444). Atmospheric profiles of temperature and humidity should be taken from radio soundings, as they provide a reliable description of the atmospheric state on the considered cloudy day. The use of longwave radiation measurements on cloudless days is not advisable because they are not representative of cloudy days. Temperature and humidity profiles have a strong effect on the longwave radiation.

3. The figures are well described but the interpretation should be extended, e.g. what causes the effects of the single radiation components.

Minor/Specific Comments

The page and line numbers refer to the pdf file: acp-2022-433-ATC1.pdf

1. The authors should be more precise in using specific terms. (i) "Radiative flux" is often wrongly used in literature. It describes a flux in units of Watt, but pyranometers and pyrgeometers measure flux densities in units of Watt per square meter. Either use the term "flux density" or, what is often used in the literature, "irradiance". (ii) Further use "surface albedo" instead of albedo. Albedo can also refer to the albedo of a cloud. (iii) Consider using the term "cloudless" or "cloud-free" instead of "clear-sky". The term "clear-sky" indicates an atmosphere that is also free of aerosols. (iV) "surface radiation anomaly" – anomaly sounds weird. Actually it is a "cloud radiative forcing", as used in literature.

2. P3L62-64: You should also mention the role of CCNs and INPs here.

3. P5L99: "surface radiation" – I would prefer to use the term "surface irradiance" and symbols in Eq. (1) something like that: $F_{net} = (F_{down} - F_{up})_{LW} + (F_{down} - F_{up})_{SW}$.

4. P8L187: What is a "regular" temperature profile?
    5. P10L231-233: Give a reason for the variability. In SW it is clearly an effect of SZA.
6. P10L239-241: What is the contribution of temperature and humidity on the differences in LW? That could be analyzed by radiative modeling.
7. P11L244-245: Spikes can be attributed to cloud edge effects, when direct fraction of the solar incident radiation and an additional diffuse contribution scattered from cloud edges falls on the radiation sensor.
8. P11L257: "The study is focuses on …" – already mentioned before. Skip it. Also details of binning were described before.
9. P14L322: Don't mention binning details.
10. P15L342-345: "Considering the SR vs. LWP relationship…" - I don't quite understand the connection with Table 3. If there is a mode that is centered around 0 W m^-2, then it probably refers to a low LWP.
11. Sec. 5.2 on the Reference Surface Radiation and sastrugi effect: Here it is not clear to me, what the benefit of using ERA5 and CERES data is for this study. Remove this section and use radiative modeling to create your cloudless references.
12. P22L513: "a lower layer being made of liquid water and an upper layer being made of solid water" – I am sure this generally true for mixed-phase clouds. To my knowledge liquid cloud particles are mainly near cloud top (https://www.pnas.org/doi/full/10.1073/pnas.1418197111).
13. P22L519-522: The positive part in the SW is probably due to cloud edge effects rather than the presence of liquid and ice particles in the cloud.
14. P22L530: "during the local "night" at 00:00-06:00 LT" – should by polar day.
15. Sec. 5.6 on the potential radiative impact of SLWCs over Antarctica: The estimation of an Antarctica-wide radiative impact of SLWCc is based on the maximum net difference. How is the effect of SZA on $F_{net}(SW)$ accounted for here, how the effect of seasonal dependent atmospheric profiles that determines the LW contribution? The 50 W m-2 is probably not a representing number for a larger area and time frame.

---

## Referee Report (RR2)

Referee comment on "Supercooled liquid water clouds observed over Dome C, Antarctica: temperature sensitivity and surface radiation impact" by Ricaud et al.

In general, the authors revised their manuscript thoroughly, taking into account all reviewers' comments. The manuscript has been significantly improved.

I only have a few minor suggestions that could be considered before the final publication.

General Comments

The authors introduced the cloud radiative forcing. Starting with the definition of the net irradiance:

$$F_{Net} = \left(F_{LW}^{Down} - F_{LW}^{Up}\right) + \left(F_{SW}^{Down} - F_{SW}^{Up}\right) \tag{1}$$

the cloud radiative forcing is difference between the net irradiances, in cloudy (Fnet, cld) and cloud-free (Fnet, cf) conditions (e.g., Stapf et al., 2020):

$\Delta F_{net,cf} = F_{net,cld} - F_{net,cf}$

The authors also refer to the difference between the individual components as cloud radiative forcing (CRF). As I understand it, radiative forcing only refers to the differences in net irradiance. Below (Specific Comments) I have listed some examples of texts that should therefore be changed.

Specific Comments

Abstract:

L37/L38: Please use either "solar" or "shortwave" throughout the manuscript.

L40 and others: "net cloud radiative forcing" – remove "net", as the CRF is related to the net irradiance.

Introduction:

L66: "Bromwich et al. (2012) mention in their review paper that CCN and INPs are of various nature and large uncertainties exist relative to their origin and abundance over Antarctica." - Do you mean variability or uncertainties?

L104: "the longwave downward" → "downward longwave", the same for the other quantities

L105: "At a given time, the impact of a cloud on the surface irradiance can be estimated by subtracting what would have been the cloud-free surface irradiance from the measured surface irradiance, to provide the so called "cloud radiative forcing"." – maybe you could write "At a given time, the impact of a cloud on the surface irradiance is estimated from the difference between the net irradiances, in cloudy ($F_{net,\ cld}$) and cloud-free ($F_{net,\ cf}$) conditions to provide the so-called ..." Why is the equation for the CRF not already given here?

Section 3.1:
L201: "The same method is used for F. BSRN Fs are time interpolated to be coincident with the other parameters." I would delete the first sentence, since the irradiance is only interpolated in time and not in space (vertical direction like the temperature). "BSRN Fs" → "BSRN irradiances"

Section 3.2:

L223-L226: "The cloud radiative forcing (ΔF) can be defined as: …": see general comments

L226: "Several studies have been performed …" – give references.

L243-L246: Please note that the surface albedo of snow under cloudy conditions may differ from the surface albedo under cloud-less conditions (e.g., Gardner and Sharp, 2010, Stapf et al., 2020). Maybe mention it here since it is another source of uncertainty.

L249: "Note that computationally simple, theoretically based parameterization for the broadband albedo of snow and ice can accurately reproduce the theoretical broadband albedo under a wide range of snow, ice, and atmospheric conditions (Gardner and Sharp, 2010)." – Why is this mentioned here? The albedo is not parameterized in this study.

L256: "Screen-level temperatures are provided by the American automated weather station (AWS) situated at ~500 m from the Concordia base." – Can be removed. It was already mentioned before.

Section 4.2:

L358-L359: "PDs of the cloud radiative forcing ΔF as a function of the LWP, for …" – CRF is $\Delta F_{net}$, the others are only components that contribute to the CRF (see general comments)

Section 5.1:

The section needs a little more interpretation. What is the critical temperature exactly? What does it tell us for this study?

L406: "SR anomaly" – you mean the CRF here, I guess

Section 5.5:

Figure 10 is not really needed.

L517-L518: "The large diurnal signal present in the observed surface albedo is likely the signature of the sastrugi effect." – It depends on sastrugi orientation (geometry) and sun geometry that affects the surface albedo. Even with a flat snow surface, one would expect the surface albedo to depend on the SZA (Gardner and Sharp, 2010). You might mention that.

L525: "We can state that the sastrugi effect on the observed cloud-free surface albedo at Concordia is successfully fitted by two sine functions of 24h and 12h periods …" – Since the orientation of sastrugis could be different, would it be possible that your fit is more related to the SZA effect?

Section 5.6:

L545: Eq. (18) assumes a linear dependence between cloud fraction and CRF. Perhaps it should be mentioned that there are 3D radiation effects in nature that contradict this assumption.

Technical comments

L285: "increases to values of +40-90 W m-2" – Here and elsewhere better write "+40 to 90 W m-2"
L551: "… over Antarctica of about 12, 10 and 7 W m-2, respectively." → 12 W m-2, 10 W m-2, and 7 W m-2

References:

Gardner, A.S. and Sharp, M.J.: A review of snow and ice albedo and the development of a new physically based broadband albedo parameterization. Journal of Geophysical Research: Earth Surface, 115(F1), 2010.

Stapf, J., Ehrlich, A., Jäkel, E., Lüpkes, C., and Wendisch, M.: Reassessment of shortwave surface cloud radiative forcing in the Arctic: consideration of surface-albedo–cloud interactions, Atmos. Chem. Phys., 20, 9895–9914, https://doi.org/10.5194/acp-20-9895-2020, 2020.

---

## Author Response (AR2)

**Revision R02 Version 04, 7 November 2023**

**Manuscript Title:** *Supercooled liquid water clouds observed over Dome C, Antarctica: temperature sensitivity and surface radiation impact* **by Ricaud et al.**

**RESPONSES TO THE EDITOR**

Dear Dr. Ricaud,

Thank you for your extensive revisions to the paper. I invited the original reviewers, as well as to an additional expert for their review of the paper. They are all in agreement that the paper is potentially highly valuable to the community, but that it is crucial in this case to use a radiative transfer model to validate your work. I strongly encourage you to address these concerns. To give you ample to time, I will reconsider after major revisions.

$\rightarrow$ Dear editor, we have modified the section relative to the estimation of the cloud-free irradiance. It took some time to converge toward the most appropriate solution. We believe the revised version now meets the reviewers' concerns.

In the Acknowledgements, we have updated the number of reviewers (from two to three):

> *We would like to thank the three anonymous reviewers for their beneficial comments.*

Also note, to be consistent with the reviewers' comments, we have modified the title from:

> *Supercooled liquid water clouds observed over Dome C, Antarctica: temperature sensitivity and surface radiation impact*

to

> *Supercooled liquid water clouds observed over Dome C, Antarctica: temperature sensitivity and radiative forcing*

**Report #1**
**Reviewer2**

→ Specific changes have been made in response to the reviewer's comments and are described below. The reviewer's comments are recalled in blue.

I thank the authors for updating their manuscript and their extensive response. However, I must admit that I am even more confused than before. I still think that simply ignoring the diurnal cycle when estimating reference SR by a mean value cannot be correct. I understand that ERA5 cannot handle sastrugi effects, but what is going on with ERA5 so that LW down is wrong by 30 W/m2 for a simple clear sky case? Or is that a measurement problem? Maybe I am also overlooking something here. This looks to me like this paper needs a reviewer who is an absolute expert in SW/LW radiation, my expertise is more in the microwave range. Therefore, I do not feel qualified to give a recommendation here (even though the system forces me to select a box).

→ Thank you for your valuable comments. We have removed the study based on ERA5 and CERES data. This is explained in detail in the responses of the reviewer3's comments.

One more minor comment: please use T instead of $\Theta$ for temperature, the latter is usually used for potential temperature, so the use of $\Theta$ is very confusing.

→ Done, "$\theta$" has been changed into "T".

**Anonymous Referee #3**

Referee comment on "Supercooled liquid water clouds observed over Dome C, Antarctica: temperature sensitivity and surface radiation impact" by Ricaud et al.

→ Specific changes have been made in response to the reviewer's comments and are described below. The reviewer's comments are recalled in blue.

The study investigates the relation between temperature and liquid water path in the presence of supercooled liquid water clouds, and analyzed the cloud radiative effect in terms of net surface radiation.

In general, the authors have attempted to thoroughly revise their manuscript according to the reviewers' comments. The relationship between temperature and LWP is well described. However, I have still some concerns regarding the methodology used to estimate the cloud radiative effect. Overall, I cannot recommend publication without a major revision.

→ Thank you for your valuable comments. We explained below the new methodology employed.

Also note, to be consistent with your comments, we have modified the title from:

*Supercooled liquid water clouds observed over Dome C, Antarctica: temperature sensitivity and surface radiation impact*

to

*Supercooled liquid water clouds observed over Dome C, Antarctica: temperature sensitivity and radiative forcing*

**Major Comments**

1. In the revised version, the authors included ERA5 and CERES data to evaluate the surface radiation in cloudless conditions. I don't think this significantly supports the study for several reasons. First, a potential effect of surface roughness (sastrugi) on surface albedo could be directly estimated from the pyranometer measurements. Second, it is not clear to me, for what reason ERA5 and CERES data were introduced here. It is known, that these data sets are not very reliable in representing the surface albedo, as also shown by the authors. Both data sets were fudged to the measurements by using the measured surface albedo for scaling the upward irradiance. What is to be done with the fitted results? How does it help to estimate the radiative effect of clouds based on the ERA5/CERES data, since they are still based on the near-time measured albedo? How do they prove the stability of conditions?

→ As suggested by the reviewer, we have definitely suppressed the use of ERA5 and CERES from the analysis and worked on the methodology proposed by the reviewer (see our response to the next comment).

2. A more feasible approach was already suggested by the previous reviewers. I strongly encourage the authors to use radiative transfer modeling to estimate the cloudless reference

surface radiation. Since the surface albedo are available directly from the measurements in cloudless conditions, even the diurnal pattern can be considered. Note, that snow/white ice albedo is also dependent on the solar zenith angle (e.g., Gardner and Sharp, 2010 - https://doi.org/10.1029/2009JF001444). Atmospheric profiles of temperature and humidity should be taken from radio soundings, as they provide a reliable description of the atmospheric state on the considered cloudy day. The use of longwave radiation measurements on cloudless days is not advisable because they are not representative of cloudy days. Temperature and humidity profiles have a strong effect on the longwave radiation.

$\rightarrow$ We have taken some time to consider using radiative transfer models to validate our work. We have used the RRTM code (Mlawer et al., 1997) and performed clear and clean sky computations. As an input, we have used the following data:

- Temperature and specific humidity profiles from the radiosondes launched every day at 12:00 UTC from the Concordia station and assumed to be constant along each day,
- ERA5 ozone profiles,
- Yearly-mean greenhouse gas concentration from NOAA, assuming these gases are well-mixed,
- Surface temperature diagnosed from the BSRN surface longwave fluxes crudely assuming a surface emissivity of 0.99,
- Surface albedo diagnosed from the BSRN surface shortwave fluxes,
- Solar zenith angle based on the calculation performed within the ARPEGE atmospheric model used for operational weather forecasting at Météo-France.

Mlawer, E.J., Taubman, S.J., Brown, P.D., Iacono, M.J. and Clough, S.A.: Radiative transfer for inhomogeneous atmospheres: RRTM, a validated correlated-k model for the longwave. Journal of Geophysical Research: Atmospheres, 102(D14), 16663-16682, https://doi.org/10.1029/97JD00237, 1997.

We have performed calculations on the 5 cloud-free 24-hour periods available (2018-12-02, 2018-12-19, 2021-12-03, 2021-12-17, 2021-12-26) and calculated the root-mean square error (RMSE). Although the RMSE scores of the difference between surface irradiances observed by the BSRN instruments and the calculations in cloud-free conditions were acceptable for upward and downward shortwave (SW) surface irradiances with values less than 10 W m$^{-2}$, and upward longwave (LW) surface irradiances with values less than 1 W m$^{-2}$, they were not acceptable for the downward longwave surface irradiances with values ranging from ~10 W m$^{-2}$ in 2018 to ~30 W m$^{-2}$ in 2021. Similar calculations were done using ERA5 reanalyses or ARPEGE analyses to prescribe the time-evolving profiles of temperature and specific humidity. These two sensitivity tests only marginally improved the results.

We thus concluded that performing radiative transfer calculation was not the appropriate solution for our objectives, at least given the data to which we have access. Going back to the literature, we have found a more relevant option, that has been applied to several other site measurements to diagnose cloud-free surface fluxes. This approach is discussed in the following.

Before explaining our methodology to estimate the cloud radiative forcing, we have to mention that, according to the minor comments of the reviewer, we have changed the spelling and terminology of the "surface irradiance" terms in eq (1) as:

$$F_{Net} = \left(F_{LW}^{Down} - F_{LW}^{Up}\right) + \left(F_{SW}^{Down} - F_{SW}^{Up}\right) \tag{1}$$

where $F_{Net}$, $F_{LW}^{Down}$, $F_{LW}^{Up}$, $F_{SW}^{Down}$, and $F_{SW}^{Up}$ represent the net, longwave downward, longwave upward, shortwave downward and shortwave upward surface irradiances, respectively. And we now write the "cloud radiative forcing" as $\Delta F$:

$$\Delta F = F - FCF \tag{2}$$

for each term of equation (1), with $FCF$ being the irradiance in cloud-free conditions.

Since radiative transfer modeling was unsatisfactory, to compute cloud-free surface irradiance in the LW and SW ranges, we follow the work of Dupont et al. (2008) and Dutton et al. (2004), respectively. Namely, in Dutton et al. (2004), cloud-free downward shortwave surface irradiance ($FCF_{SW}^{Down}$) is parameterized as:

$$FCF_{SW}^{Down} = a \, \cos(z)^b \, c^{\left(\frac{-1}{\cos(z)}\right)} \tag{3}$$

where $z$ is the solar-zenith angle, and $a$, $b$, and $c$ are coefficients optimized using well-identified cloud-free situations. In Dupont et al. (2008), cloud-free downward longwave surface irradiance ($FCF_{LW}^{Down}$) is parameterized as:

$$FCF_{LW}^{Down} = \varepsilon_a \, \sigma \, T_a^4 \tag{4}$$

where $T_a$ is the screen-level air temperature in Kelvin (K), $\sigma$ the Stephan-Boltzmann's constant and $\varepsilon_a$ the apparent atmospheric emissivity. The latter is supposed to be a function of the integrated water vapor (IWV) following the equation:

$$\varepsilon_a = 1 - (1 + IWV)\exp(-(d + e \times IWV)^f) \tag{5}$$

where $d$, $e$ and $f$ are coefficients that need to be optimized using cloud-free situations. The cloud-free upward shortwave surface irradiance ($FCF_{SW}^{Up}$) is evaluated from $FCF_{SW}^{Down}$ with the surface albedo ($A_{BSRN} = F_{SW}^{Up}(BSRN)/F_{SW}^{Down}(BSRN)$) calculated from observations:

$$FCF_{SW}^{Up} = A_{BSRN} \times FCF_{SW}^{Down} \tag{6}$$

where $F_{SW}^{Up}(BSRN)$ and $F_{SW}^{Down}(BSRN)$ are the upward and downward shortwave surface irradiances measured by the BSRN instruments, respectively. With this method, we are able to take into account the diurnal variability of the surface albedo which is dependent on the sun angles because of the sastrugi effect present at Concordia. Note that a computationally simple, theoretically based parameterization for the broadband albedo of snow and ice can accurately reproduce the theoretical broadband albedo under a wide range of snow, ice, and atmospheric conditions (Gardner and Sharp, 2010).

The cloud-free upward longwave radiation ($FCF_{LW}^{Up}$) is evaluated as:

$$FCF_{LW}^{Up} = \varepsilon_s \, \sigma \, T_s^4 + (1 - \varepsilon_s) \, FCF_{LW}^{Down} \tag{7}$$

where $T_s$ is the surface temperature (in K) and the surface emissivity $\varepsilon_s$ is assumed constant and equal to 0.99.

[revised manuscript text omitted]

$$a = 1360.7 \ [1360.5, 1360.8] \ \mathrm{W \ m^{-2}}$$
$$b = 0.990 \ [0.989, 0.991]$$
$$c = 0.964 \ [0.964, 0.965]$$
$$bias = -0.002 \ [-0.317, 0.251] \ \mathrm{W \ m^{-2}}$$
$$RMSE = 14.9 \ [10.8, 16.5] \ \mathrm{W \ m^{-2}}$$

Similarly, for downward longwave surface irradiance, the K-fold cross-validation provides the following results:

$$d = 0.723 \ [0.722, 0.724]$$
$$e = 3.58 \ [3.57, 3.59] \ \mathrm{kg^{-1} \ m^2}$$
$$f = 1.0$$
$$bias = 0.34 \ [-0.005, 0.87] \ \mathrm{W \ m^{-2}}$$
$$RMSE = 9.26 \ [8.92, 9.58] \ \mathrm{W \ m^{-2}}$$

These coefficient values are used to compute cloud-free surface irradiances at a 1-min time resolution.

M. A. Branch, T. F. Coleman, and Y. Li, "A Subspace, Interior, and Conjugate Gradient Method for Large-Scale Bound-Constrained Minimization Problems," SIAM Journal on Scientific Computing, Vol. 21, Number 1, pp 1-23, 1999.

Based on these cloud-free calculations, we can now estimate the impact of the SLW clouds on the different components of the cloud radiative forcing. As an example, Figures R1 and R2 show the cloud radiative forcing $\Delta F$ on 9 December 2019 and 27 December 2021, respectively.

[Figure]

**Figure R1:** Time evolution (UTC, hour) of the cloud radiative forcing components ($\Delta F$) (W m$^{-2}$) calculated on 9 December 2019: (from top to bottom) net, longwave downward, longwave upward, shortwave downward and shortwave upward. The SLW cloud layer (if present) is

highlighted by a red area in the uppermost panel, with the height on the y-axis shown on the right. The 00:00 and 12:00 local times (LT) are highlighted by 2 vertical blue dashed lines.

[Figure]

**Figure R2:** Same as Figure R1 but on 27 December 2021.

Now, if we consider the complete data set encompassing the 4 Decembers 2018-2021, the relationship between $LWP$ (g m$^{-2}$) and $\Delta F_{net}$ (W m$^{-2}$) as seen in Figure R3 was shifted with respect to the estimates presented in the previous version of our paper (Figure R4). We note that there is no longer negative forcing irradiance (significant $\Delta F_{net}$ values are positive whatever the positive $LPW$ values) which is now expressed as:

$$\Delta F_{net} = (-18 \pm 10) + 70 \ln(LWP) \qquad (8)$$

[Figure]

**Figure R3:** (Left) Probability Density (PD, %) of the net cloud radiative forcing ($\Delta F_{net}$, W m$^{-2}$) as a function of Liquid Water Path (LWP, g m$^{-2}$) contained in the Supercooled Liquid Water clouds (SLWCs) above Dome C in December 2018-2021. The PD is defined in the manuscript. (Centre) Weighted-average LWP vs. $\Delta F_{net}$ with a fitted logarithmic function (blue solid) encompassing the significant points (2 dashed blue lines). Horizontal bars represent 1-sigma variability in LWP per 5 W m$^{-2}$-wide bin over significant points. (Right) $\Delta F_{net}$ as a function of count number per 5 W m$^{-2}$-wide bin (black solid line) with 3 fitted Gaussian functions (red dashed curves). The sum of the 3 Gaussian functions is represented by a red solid line.

[Figure]

**Figure R4:** Figure presented in the very first version of the paper showing negative bias in $\Delta F_{net}$.

And we have reproduced all the remaining Figures (R5-8) showing Probability Density (PD, %) of the cloud radiative forcing in the LW and SW downward and upward (W m$^{-2}$) as a function of Liquid Water Path (LWP, g m$^{-2}$) contained in the Supercooled Liquid Water clouds (SLWCs) above Dome C in December 2018-2021.

December 2018-2021

[Figure]

**Figure R5:** Same as Figure R3 but for $\Delta F_{LW}^{Down}$.

December 2018-2021

[Figure]

**Figure R6:** Same as Figure R3 but for $\Delta F_{LW}^{Up}$.

[Figure]

**Figure R7:** Same as Figure R3 but for $\Delta F_{SW}^{Down}$.

[Figure]

**Figure R8:** Same as Figure R3 but for $\Delta F_{SW}^{Up}$.

For each $T$ and $\Delta F$ dataset, the distribution of the total count numbers $N_{tj}$ per 1°C or 5 W m$^{-2}$-wide bin ($N_{tj} = \sum_{i=1}^{M} N_{ij}$ with $j = 1, ..., N$) can be fitted by a function $N(x)$, with $x = T$ or $\Delta F$, based on 2 to 3 Gaussian distributions as:

$$N(x) = \sum_{k=1}^{2 \text{ or } 3} a_k \exp\left(-\frac{1}{2}\left(\frac{x-\mu_k}{\sigma_k}\right)^2\right) + c \qquad (9)$$

with $a_k$, $\mu_k$ and $\sigma_k$ being the amplitude, the mean and the standard deviation of the $k^{th}$ Gaussian function ($k = 1, 2$ or $3$) and $c$ is a constant. Table R2 lists all the fitted parameters ($a_k$, $\mu_k$, $\sigma_k$ and $c$ with $k = 1$ to 2 or 3).

**Table R2.** Gaussian functions fitted to the $N(x)$ function for $x = T$ (°C) or $\Delta F$ (W m$^{-2}$). Units of $a_1$, $a_2$, $a_3$, and $c$ are in count number for $T$ and $\Delta F$; units of $\mu_1$, $\mu_2$, $\mu_3$, $\sigma_1$, $\sigma_2$, and $\sigma_3$ are in °C for $T$ and in W m$^{-2}$ for $\Delta F$.

| $x$ | $a_1$ | $\mu_1$ | $\sigma_1$ | $a_2$ | $\mu_2$ | $\sigma_2$ | $a_3$ | $\mu_3$ | $\sigma_3$ | $c$ |
|---|---|---|---|---|---|---|---|---|---|---|
| $T$ | $15.0\ 10^3$ | -31.5 | 1.45 | $5.0\ 10^3$ | -28.0 | 1.65 | $0.5\ 10^3$ | -19.0 | 2.5 | $-9.1\ 10^{-6}$ |
| $\Delta F_{net}$ | 371.7 | 10.0 | 11.5 | 74.6 | 37.6 | 21.1 | 220.8 | 57.5 | 14.1 | -10.2 |
| $\Delta F_{LW}^{Down}$ | 415.5 | 10.0 | 10.4 | 189.5 | 53.7 | 24.2 | 227.1 | 82.9 | 7.0 | -18.5 |
| $\Delta F_{LW}^{Up}$ | - | - | - | - | - | - | - | - | - | - |
| $\Delta F_{SW}^{Down}$ | 190.5 | -10.1 | 17.2 | 113.0 | -80.0 | 54.6 | - | - | - | -1.9 |
| $\Delta F_{SW}^{Up}$ | 282.4 | -10.1 | 12.8 | 133.8 | -75.0 | 41.8 | - | - | - | 8.3 |

And a new logarithmic function of the form

$$x = \alpha + \beta \ln(\overline{LWP}) \qquad (10)$$

has been fitted onto these significant points where the retrieved constants $\alpha$ and $\beta$ are shown in Table R3.

**Table R3.** Coefficients of the relations $f(LWP) = \alpha + \beta \ln(LWP)$ for the temperature $T$ or cloud radiative forcing $\Delta F$. Units of $T$ and $\Delta F$, as well as of their corresponding "$\alpha$" values are in °C and W m$^{-2}$, respectively; units of $\beta$ are in °C g$^{-1}$ m$^2$ for $T$ and in W g$^{-1}$ for $\Delta F$; units of LWP are in g m$^{-2}$. The last column shows the range of LWP values for which the relation is valid. Note that $\alpha \pm \delta\alpha$ corresponds to the range of $\alpha$ values where the relationship is valid.

| $f(LWP)$ | $\alpha \pm \delta\alpha$ | $\beta$ | Valid range for $T$ or $\Delta F$ | Valid range for LWP |
|---|---|---|---|---|
| $T$ | $-33.8 \pm 1.5$ | 6.5 | $[-36; -16]$ | $[1.0; 14.0]$ |
| $\Delta F_{net}$ | $-18.0 \pm 10.0$ | 70.0 | $[0; 70]$ | $[1.2; 3.5]$ |
| $\Delta F_{LW}^{Down}$ | $5.0 \pm 15.0$ | 65.0 | $[0; 90]$ | $[1.0; 3.5]$ |
| $\Delta F_{LW}^{Up}$ | $0 \pm 5.0$ | 0.0 | $[-5; 5]$ | $[0.0; 6.5]$ |
| $\Delta F_{SW}^{Down}$ | $30.0 \pm 30.0$ | -130.0 | $[-130; 0]$ | $[1.5; 4.0]$ |
| $\Delta F_{SW}^{Up}$ | $30.0 \pm 30.0$ | -110.0 | $[-110; 00]$ | $[1.5; 4.0]$ |

3. The figures are well described but the interpretation should be extended, e.g. what causes the effects of the single radiation components.

→ To summarize, the impact of SLWCs on single radiation components is:

- LW down: main term for the cloud-related impact. Order of magnitude: 0-80 W m$^{-2}$
- LW up: marginal impact since there is no relation with LWP, it is mainly dependent on $T_s$ which results from various meteorological forcing
- SW down: a huge reduction of direct solar radiation, which can overpass 100 W m$^{-2}$, but is largely compensated for by:
- SW up: the very high surface albedo values result in a weak net SW radiation.

In conclusion, the driving term is the downward longwave irradiance.

**Minor/Specific Comments**
The page and line numbers refer to the pdf file: acp-2022-433-ATC1.pdf

1. The authors should be more precise in using specific terms. (i) "Radiative flux" is often wrongly used in literature. It describes a flux in units of Watt, but pyranometers and pyrgeometers measure flux densities in units of Watt per square meter. Either use the term "flux density" or, what is often used in the literature, "irradiance". (ii) Further use "surface albedo" instead of albedo. Albedo can also refer to the albedo of a cloud. (iii) Consider using the term "cloudless" or "cloud-free" instead of "clear-sky". The term "clear-sky" indicates an atmosphere that is also free of aerosols. (iV) "surface radiation anomaly" – anomaly sounds weird. Actually it is a "cloud radiative forcing", as used in literature.

→ In the revised version, we have modified all the terms as recommended by the reviewers into: "cloud-free", "irradiance", "surface albedo" and "cloud radiative forcing". See above "Main Comments, section 2"

2. P3L62-64: You should also mention the role of CCNs and INPs here.

→ We have inserted some information relative to CCNs and INPs.

The abundance of Supercooled Liquid Water (SLW, the water staying in liquid phase below 0°C) clouds depends on temperature and liquid/ice fraction. Furthermore, the nature and optical properties of the clouds depend on the type and concentration of cloud condensation nuclei (CCNn) and ice-nucleating particles (INPs). Bromwich et al. (2012) mention in their review paper that CCNs and INPs are of various nature and large uncertainties exist relative to their origin and abundance over Antarctica.

3. P5L99: "surface radiation" – I would prefer to use the term "surface irradiance" and symbols in Eq. (1) something like that: F_net = (F_down-F_up)_LW + (F_down-F_up)_SW.

→ We modified the terms, see above "Main Comments, section 2"

4. P8L187: What is a "regular" temperature profile?

→ We changed the term into "temperature profile".

5. P10L231-233: Give a reason for the variability. In SW it is clearly an effect of SZA.

→ The sentence referred to here was relative to a sometimes positive and sometimes negative cloud impact on the net radiation in the previous manuscript. With the new calculations presented in the revised version, we can see that the impact is now almost always positive (see Fig. R3), which seems consistent.

6. P10L239-241: What is the contribution of temperature and humidity on the differences in LW? That could be analyzed by radiative modeling.

→ To answer this question, it is necessary first to look at the values of the equivalent atmospheric emissivity $\varepsilon_a$ used in the equations 4-7 above. The values of Integrated Water Vapour (IWV) observed at Dome C are very low even in summer, typical summertime values are between 0.8 and 1.2 kg m$^{-2}$ (Ricaud et al., 2020). This corresponds to values of $\varepsilon_a$ between 0.950 and 0.985, i.e. a relative variation of the order of 3.6%. On the other side, a variation $\Delta T$ of the screen-level (surface) temperature $T_a$ ($T_s$) has a relative impact the downwelling (upwelling) longwave irradiance of the order of $4\, \Delta T/T_a$ ($4\, \Delta T/T_s$), which amounts to around 1.6% per degree of $\Delta T$. Given that observations of surface and screen-level temperatures reveal variations of several degrees, both in their diurnal cycle and from a day to another, we can conclude that the impact of temperature on longwave irradiance variations is larger than that of IWV.

We have thus inserted a new paragraph in the revised version that synthetizes our results.

7. P11L244-245: Spikes can be attributed to cloud edge effects, when direct fraction of the solar incident radiation and an additional diffuse contribution scattered from cloud edges falls on the radiation sensor.

→ This point is included in the new version of the manuscript.

8. P11L257: "The study is focuses on …" – already mentioned before. Skip it. Also details of binning were described before.

→ Removed.

9. P14L322: Don't mention binning details.

→ Removed.

10. P15L342-345: "Considering the SR vs. LWP relationship…" - I don't quite understand the connection with Table 3. If there is a mode that is centered around 0 W m^-2, then it probably refers to a low LWP.

→ Removed.

11. Sec. 5.2 on the Reference Surface Radiation and sastrugi effect: Here it is not clear to me, what the benefit of using ERA5 and CERES data is for this study. Remove this section and use radiative modeling to create your cloudless references.

→ Section removed regarding ERA5 and CERES, but we kept the discussion on the sastrugi effects on the surface albedo since it is a key component to characterize the solar upward irradiance.

12. P22L513: "a lower layer being made of liquid water and an upper layer being made of solid water" – I am sure this generally true for mixed-phase clouds. To my knowledge liquid cloud particles are mainly near cloud top (https://www.pnas.org/doi/full/10.1073/pnas.1418197111).

→ From literature (see e.g. Fig. 12.3 from Lamb and Verlinde (2011)) and LIDAR observations at Concordia (see e.g. Figure 1 of the present paper), liquid (low depolarization ratio) is below solid (high depolarization ratio) water in cold clouds above Concordia.

13. P22L519-522: The positive part in the SW is probably due to cloud edge effects rather than the presence of liquid and ice particles in the cloud.

→ Positive part is not significant in the new study.

14. P22L530: "during the local "night" at 00:00-06:00 LT" – should by polar day.

→ Yes. We modified the term into:

"at 00:00-06:00 LT when the sun is at low elevation above the horizon (24-h polar day),"

15. Sec. 5.6 on the potential radiative impact of SLWCs over Antarctica: The estimation of an Antarctica-wide radiative impact of SLWCc is based on the maximum net difference. How is the effect of SZA on F_net(SW) accounted for here, how the effect of seasonal dependent atmospheric profiles that determines the LW contribution? The 50 W m-2 is probably not a representing number for a larger area and time frame.

→ We have re-analyzed the results by considering a maximum of 70 W m$^{-2}$ in the net cloud radiative impact (present analysis) that gives a slightly bigger impact compared to our previous analysis at the scale of the Antarctic continent. Our analysis does not require vertical profiles of temperature. Based on our study performed locally at Concordia in summer, we can estimate that, over the Antarctic continent, SLWCs have a maximum net radiative forcing rather weak over the Eastern Antarctic Plateau (0-7 W m$^{-2}$) but 3 to 5 times larger over Western Antarctica (0-40 W m$^{-2}$) maximizing in summer over the Antarctic Peninsula.

To conclude, this study showed that the major impact of SLWCs on net surface irradiance is an increase of downward longwave component. In the presence of SLWC, the attenuation of solar incoming radiation is almost compensated for by upward shortwave irradiance because of high values of surface albedo, whatever the solar zenithal angle. But the seasonality impact must be considered, since SLWCs are preferentially observed in summer.

---

## Author Response (AR3)

**Revision R03 Version 02, 27 November 2023**

**Manuscript Title:** *Supercooled liquid water clouds observed over Dome C, Antarctica: temperature sensitivity and radiative forcing* **by Ricaud et al.**

**RESPONSES TO THE EDITOR**

Dear Dr. Ricaud,

I am in the possession of a review of your current manuscript. Overall, the reviewer was happy with the improvements you made to the manuscript, but has a few more things to look into. This should only be a minor revision before the manuscript can be published.

→ Dear editor, thank you very much for the confidence you had in our study despite the fact that it took some time for us to converge towards fully acceptable results. We have modified the article according to the review of the last reviewer. We have updated the date of the last access to all the databases to 27 November 2023.

We also modified the title to be consistent with the reviewer's comments regarding the term cloud radiative forcing (CRF) from

[revised manuscript text omitted]

**Anonymous Referee #3**

Referee comment on "Supercooled liquid water clouds observed over Dome C, Antarctica: temperature sensitivity and surface radiation impact" by Ricaud et al.

In general, the authors revised their manuscript thoroughly, taking into account all reviewers' comments. The manuscript has been significantly improved.

I only have a few minor suggestions that could be considered before the final publication.

→ Thank you very much for your fruitful comments.

General Comments

The authors introduced the cloud radiative forcing. Starting with the definition of the net irradiance:

$$F_{Net} = \left(F_{LW}^{Down} - F_{LW}^{Up}\right) + \left(F_{SW}^{Down} - F_{SW}^{Up}\right) \qquad (1)$$

the cloud radiative forcing $\Delta F_{Net}$ is difference between the net irradiances, in cloudy ($F_{Net,cld}$) and cloud-free ($F_{Net,cf}$) conditions (e.g., Stapf et al., 2020):

$$\Delta F_{Net} = F_{Net,cld} - F_{Net,cf} \qquad (2)$$

The authors also refer to the difference between the individual components as cloud radiative forcing (CRF). As I understand it, radiative forcing only refers to the differences in net irradiance.

→ We understand this key point. We have modified the text accordingly throughout the manuscript. We now refer to CRF only considering the net irradiances, and how CRF is split into the individual components ($F_{LW}^{Down}$, $F_{LW}^{Up}$, $F_{SW}^{Down}$ and $F_{SW}^{Up}$).

Below (Specific Comments) I have listed some examples of texts that should therefore be changed.

Specific Comments

Abstract:

L37/L38: Please use either "solar" or "shortwave" throughout the manuscript.

→ We now only use "shortwave" throughout the manuscript.

L40 and others: "net cloud radiative forcing" – remove "net", as the CRF is related to the net irradiance.

→ Yes see point above.

Introduction:

L66: "Bromwich et al. (2012) mention in their review paper that CCN and INPs are of various nature and large uncertainties exist relative to their origin and abundance over Antarctica." - Do you mean variability or uncertainties?

→ We are actually referring to "uncertainties" according to the first sentence of abstract in Bromwich et al. (2012): "*Compared to other regions, little is known about clouds in Antarctica.*"; and to the last sentence of this abstract "*While cloud monitoring over Antarctica from space has proved essential to the recent advances, the review concludes by emphasizing the need for additional in situ measurements.*"

L104: "the longwave downward" → "downward longwave", the same for the other quantities

→ Done throughout the manuscript.

L105: "At a given time, the impact of a cloud on the surface irradiance can be estimated by subtracting what would have been the cloud-free surface irradiance from the measured surface irradiance, to provide the so called "cloud radiative forcing"." – maybe you could write "At a given time, the impact of a cloud on the surface irradiance is estimated from the difference between the net irradiances, in cloudy ($F_{Net,cld}$) and cloud-free ($F_{Net,cf}$) conditions to provide the so-called ..." Why is the equation for the CRF not already given here?

→ We have modified the sentence and introduced the cloud radiative forcing there, including the relevant equation.

Section 3.1:

L201: "The same method is used for F. BSRN Fs are time interpolated to be coincident with the other parameters." I would delete the first sentence, since the irradiance is only interpolated in time and not in space (vertical direction like the temperature). "BSRN Fs" → "BSRN irradiances"

→ We have modified the text according to the reviewer's comments.

Section 3.2:

L223-L226: "The cloud radiative forcing (ΔF) can be defined as: …": see general comments

→ The issue of "cloud radiative forcing" has been carefully dealt throughout the manuscript.

L226: "Several studies have been performed …" – give references.

→ There is a misunderstanding in this sentence since we were referring to our own studies. We modified the sentence into:

We performed several studies (reference irradiances measured over days when clouds are absent, radiative transfer calculations) from which it resulted that the most robust method was to use a parameterization of the cloud-free downward longwave and shortwave surface irradiances widely used in the community.

L243-L246: Please note that the surface albedo of snow under cloudy conditions may differ from the surface albedo under cloud-less conditions (e.g., Gardner and Sharp, 2010, Stapf et al., 2020). Maybe mention it here since it is another source of uncertainty.

→ We have inserted the sentence proposed by the reviewer in the section "5.5. Sastrugi effect on the surface albedo" and included the two references.

L249: "Note that computationally simple, theoretically based parameterization for the broadband albedo of snow and ice can accurately reproduce the theoretical broadband albedo under a wide range of snow, ice, and atmospheric conditions (Gardner and Sharp, 2010)." – Why is this mentioned here? The albedo is not parameterized in this study.

→ We removed this sentence.

L256: "Screen-level temperatures are provided by the American automated weather station (AWS) situated at ~500 m from the Concordia base." – Can be removed. It was already mentioned before.

→ Removed.

Section 4.2:

L358-L359: "PDs of the cloud radiative forcing ΔF as a function of the LWP, for …" – CRF is ΔFnet, the others are only components that contribute to the CRF (see general comments)

→ Modified according to the reviewer's comments.

Section 5.1:

The section needs a little more interpretation. What is the critical temperature exactly? What does it tell us for this study?

→ We understand the point. Our (naïve) approach was simply to highlight that the exponential relationship between supercooled liquid water and in-cloud temperature we have observed in our study can be related to the theory that also shows this exponential behavior (see e.g. Figure 4 from Sippola and Taskinen (2018)) for temperatures less than 0°C.

[Figure]

Figure 4. Assessed density ρ of liquid water compared with experimental data.[15,23,33,34] Dotted lines indicate extrapolated values. Data by Speedy[15] and Lind and Trusler[34] were not included in the assessment.

Figure taken from Sippola and Taskinen (2018).

In thermodynamics, a critical point (or critical state) is the end point of a phase equilibrium curve. In our study of supercooled liquid water dependence in temperature, the liquid–ice boundary terminates in an endpoint at some critical temperature Tc. Tc is about 224.8 K if water is pure and free of nucleation nuclei. But Sippola and Taskinen (2018) reviewed a value of Tc ~227-228 K (approx. -45°C) in the literature. Our study shows that, above Concordia, we could not observe SLWCs at temperatures less than -36°C that is consistent with the fact that the threshold temperature should be around -39°C (see the discussions on errors in section 5.3).

We have modified the text as follows (and changed the labelling of equation 18 into 17 since we removed one equation in this subsection).

Our study shows that, above Concordia, there is an exponential dependence of LWP on both temperature and cloud radiative forcing, that is to say supercooled liquid water exponentially increases with temperature in the range -36°C to -16°C. This is in agreement with the outputs from a simple model for thermodynamic properties of water from sub-zero temperatures up to +100°C (Sippola and Taskinen, 2018). The model shows that the density $\rho$ (g cm$^{-3}$) of liquid water exponentially increases with temperature from -34°C to 0°C through the following relationship:

$$\rho = \rho_0 \exp\{-T_c(A + B\varepsilon_0 + 2C\varepsilon_0^{1/2})\} \tag{15}$$

where $\rho_0$ = 1.007853 g cm$^{-3}$, $A$ = 3.9744 10$^{-4}$ K$^{-1}$, $B$ = 1.6785 10$^{-3}$ K$^{-1}$, and $C$ = -7.8165 10$^{-4}$ K$^{-1}$ are parameters; $T_c$ is the critical temperature (K) and $\varepsilon_0$ (unitless) is defined as:

$$\varepsilon_0 = \frac{T}{T_c} - 1 \tag{16}$$

where $T$ is temperature in K. In thermodynamics, a critical point is the end point of a phase equilibrium curve. In our study, the liquid–ice boundary terminates at some critical temperature $T_c$. $T_c$ is about 224.8 K if water is pure and free of nucleation nuclei. Sippola and Taskinen (2018) reviewed a value of $T_c$ ~227-228 K (approx. -45°C) in the literature. This is also in agreement with the results from our study showing that, above Concordia, we could not observed SLWCs at temperatures less than -36°C consistent with the fact that the threshold temperature to get SLWCs should be around -39°C (see the discussions on errors in section 5.3).

L406: "SR anomaly" – you mean the CRF here, I guess

→ Yes, it is a remnant of the very first version of the paper. We modified the term.

Section 5.5:

Figure 10 is not really needed.

→ We have removed this Figure and modified the Figure numbering accordingly.

L517-L518: "The large diurnal signal present in the observed surface albedo is likely the signature of the sastrugi effect." – It depends on sastrugi orientation (geometry) and sun geometry that affects the surface albedo. Even with a flat snow surface, one would expect the surface albedo to depend on the SZA (Gardner and Sharp, 2010). You might mention that.

→ We modified the sentence into:

"The large diurnal signal present in the observed surface albedo is likely the signature of 1) the sastrugi orientation and also 2) the sun zenith angle which impacts on the surface albedo even with a flat snow surface (Gardner and Sharp, 2010)."

L525: "We can state that the sastrugi effect on the observed cloud-free surface albedo at Concordia is successfully fitted by two sine functions of 24h and 12h periods …" – Since the orientation of sastrugis could be different, would it be possible that your fit is more related to the SZA effect?

$\rightarrow$ This is a good comment. We cannot rule out that the SZA effect also impacts on the surface albedo. Sastrugis orientation depends on the wind orientation that is climatologically blowing to the North at Concordia. Therefore, it is difficult to quantify the impact of these two effects (sastrugis and SZA) on the diurnal cycle of the surface albedo. We have inserted a new sentence:

"We cannot rule out that the diurnal cycle of the surface albedo is also impacted by the diurnal cycle of the solar zenith angle."

Section 5.6:

L545: Eq. (18) assumes a linear dependence between cloud fraction and CRF. Perhaps it should be mentioned that there are 3D radiation effects in nature that contradict this assumption.

$\rightarrow$ We have inserted a new sentence, updating "Equation (18)" into "Equation (17)".

"Equation (17) assumes a linear dependence between cloud fraction and cloud radiative forcing although, in nature, there could be three-dimensional radiation effects."

Technical comments

L285: "increases to values of +40-90 W m-2" – Here and elsewhere better write "+40 to 90 W m-2"

$\rightarrow$ Done throughout the manuscript.

L551: "… over Antarctica of about 12, 10 and 7 W m-2, respectively." $\rightarrow$ 12 W m-2, 10 W m-2, and 7 W m-2

$\rightarrow$ Done

$\rightarrow$ Reference inserted in the revised manuscript.